# Split-and-Denoise: Protect large language model inference with local differential privacy

## Abstract

Large Language Models (LLMs) show powerful capabilities in natural language understanding by capturing hidden semantics in vector space. This process enriches the value of text embeddings for various downstream tasks, thereby fostering the Embedding-as-a-Service (EaaS) business model. However, the direct transmission of text to servers poses a largely unaddressed risk of privacy leakage. To mitigate this issue, we introduce Split-N-Denoise (SnD), an innovative framework that splits the model to execute the token embedding layer on the client side at minimal computational cost. This allows the client to introduce noise prior to transmitting the embeddings to the server, and subsequently receive and denoise the perturbed output embeddings for downstream tasks. Our approach is designed for the inference stage of LLMs and requires no modifications to the model parameters. Extensive experiments demonstrate SnD's effectiveness in optimizing the privacy-utility tradeoff across various LLM architectures and diverse downstream tasks. The results reveal an improvement in performance under the same privacy budget compared to the baselines by over 10% on average, offering clients a privacy-preserving solution for local privacy protection.

## 1 Introduction

Large Language Models (LLMs) have shown powerful capability in natural language understanding by capturing hidden semantics in vector space. Consequently, users can leverage LLMs to obtain embeddings and subsequently apply them to their own downstream tasks, known as "embedding as a service" (EaaS). However, EaaS is typically provided as an online service, giving rise to significant privacy concerns. In particular, users may input sensitive information, such as names, phones, and email addresses, that needs to be kept hidden from the service provider. With the growing concern around the potential leakage of confidential data, certain companies, such as Samsung, have temporally prohibited the usage of online LLM services.

Recent research on privacy-preserving model inference investigates around two directions, cryptographic Liu & Liu (2023); Chen et al. (2022) and perturbation Du et al. (2023). Cryptography typically employs homomorphic encryption (HE) to compute the inference result of the users' encrypted input. Unfortunately, the application of cryptographic technique is constrained by the significant computation overhead of cryptographic operations, especially on large transformer models. Perturbation provides differential privacy (DP) guarantee by adding calibrated noise to the original data. A key challenge of this approach is how to balance the utility and privacy tradeoff in a local differential privacy (LDP) setting, where users' inputs are privatized before being released to the server. Furthermore, privatization on text data is particularly difficult when the randomized algorithm is required to map text input to text output.

Split learning Gupta & Raskar (2018); Vepakomma et al. (2018) has emerged as a solution to privacy-preserving computation between two parties. During inference, the user performs affordable computation locally to obtain intermediate results (IRs), and forwards them to the service provider for subsequent operations. To mitigate privacy leakage, recent research has integrate DP with split learning by injecting noises into the IRs before sharing with the server Yang et al. (2022). In the split

inference setting, a crucial problem is to design an algorithm that minimizes the impact on model performance while ensuring LDP.

A notable approach involves the application of denoising techniques to conduct error correction and enhance model utility. Existing studies incorporate denoising layers on the server side, leveraging the post-processing properties of DP Nasr et al. (2020); Wang et al. (2019); Xu et al. (2022). However, the effectiveness of denoising is hindered by the fact that the server is ignorant of the injected noise levels. Driven by the limitation, a question arises: *can we improve the utility by conducting denoising on the user side, leveraging the knowledge of noise levels and raw IRs?* It is a highly nontrivial task to unconver the closed-form mapping between denoised embedding and noises as well as raw IRs, since the inputs have undergone a series of complex transformations.

In this paper, we answer this question affirmatively by proposing Split-N-Denoise (SnD), a framework that integrates split inference and denoising techniques to enhance utility under LDP bound. To minimize computational overhead for users, we deploy only the token representation layer on the client sides. A denoise model that enhances noisy embeddings using raw inputs and noise levels is pre-trained on the server side and subsequently shared with the user. Once receiving the output from server, users input their private data into the denoise model to improve the utility of embeddings.

Our main contributions involve the following:

- We propose SnD, a framework that integrates split inference and denoising techniques to protect user's privacy during LLM inference with strong privacy guarantee. Empirical studies demonstrate that our method outperforms existing DP-based baselines by over 10% on average, and maintains utility even in extremely low privacy budget settings ($\eta \leq 0.01$).

- We design a novel denoising method deployed on user side. In this approach, a denoise model is pre-trained on server side using public dataset and synthetic noises. Subsequently, this trained model is deployed on the user side, where it leverages the specific noise levels and raw IRs provided by the user to enhance the embeddings.

## 2 PRIOR WORKS

**Local Privacy Protection for LLMs** With the advent of LLMs, privacy leakage has emerged as a crucial concern. Existing literature predominantly focuses on privacy protecti throughout the entire training process, encompassing pre-training Hoory et al. (2021), fine-tuning Huang et al. (2020); Kerrigan et al. (2020); Yu et al. (2021); Lukas et al. (2023), and prompt-tuning phases Duan et al. (2023); Li et al. (2023). Yet, there is a notable dearth of research that addresses local privacy during the inference phase with a fully frozen LLM. This scenario, which prohibits alterations to the model's structure and parameters, is particularly complex. Nonetheless, it holds significance in black-box API access contexts, especially for proprietary models like GPT-4. An intuitive approach involves anonymizing sensitive terms prior to LLM input and subsequently restoring them post-output Kan et al. (2023); Chen et al. (2023). However, this method, while effective for obfuscating specific entities, falls short in concealing other linguistic elements, including verbs and non-named entities. Such a limitation compromises full privacy and is unsuitable for tasks necessitating exact semantic interpretation of the altered entities, such as knowledge retrieval and text continuation Chen et al. (2023). An alternative strategy might entail privatizing the input at token representations or intermediate layer levels. Qu et al. (2021b) investigates the utility and privacy tradeoff for privacy-preserving finetuing, involving text-to-text privatization Feyisetan et al. (2019); Qu et al. (2021a) and token embedding privatizations, while the two techniques could be adapted to private LLM inference. Privacy-Preserving Prompt Tuning (RAPT) Li et al. (2023) employs text-text privatization to conduct prompt tuning and inference with local differential privacy. The authors propose a reconstruction head during prompt tuning to enhance the utility. Another direction employs homomorphic encryption (HE) to conduct private transformer inference such as Privacy-Computing Friendly Transformers (PCFT) and The-x Liu & Liu (2023); Chen et al. (2022), but the significant overhead renders it impractical for implementation in LLM.

**Privacy-Preserving Split Learning** Split learning is a privacy-preserving approach in distributed learning, where each client trains a segment of a deep network up to a designated "cut layer." The outputs at this layer are then forwarded to the server side, which completes the training without

accessing the client's raw data. This approach facilitates forward and backward propagation without sharing raw data, ensuring the client-side local privacy Gupta & Raskar (2018); Vepakomma et al. (2018). Vepakomma et al shows that split learning surpasses federated learning and large batch synchronous SGD in achieving superior accuracy with significantly reduced client-side computational demands Gupta & Raskar (2018). Singh et al further validate its efficacy across broader experimental contexts, demonstrating that an increase in the number of clients or model dimensions gives split learning an edge over federated learning Singh et al. (2019). The advantage in its computational efficiency renders it suitable for LLM local privacy setting, where the client side executes minimal computational tasks, such as noising and denoising operations at specific segmented layers, to ensure privacy at reduced computational expenses. Meanwhile, the server handles the bulk of the model's layers. Our research serves as an initial endeavor to integrate split learning with LLM privacy concerns.

**Denoising for Differential Privacy (DP)**   While elevated noise levels offer robust privacy protections, privacy-preserving methods inevitably compromise the model's quality Wang et al. (2019). A notable approach involves the application of denoising techniques specifically tailored for Differential Privacy (DP), incorporating a post-processing layer to enhance DP utility. Pioneering research in statistical estimation underscores the efficacy of post-processing denoising in achieving accurate private network degree distribution estimates Hay et al. (2009), and in reducing linear regression estimation errors when the ground truth is sparse Nikolov et al. (2013). Balle et al. demonstrated that denoising significantly enhances the Gaussian mechanism's accuracy in high-dimensional settings for DP algorithms with output perturbations Balle & Wang (2018). More recently, denoising mechanisms have been extended to the training of Machine Learning (ML) models, particularly Deep Neural Networks (DNNs), by applying denoising techniques to Gaussian noise-injected gradients, thereby improving the utility of privately trained ML models Wang et al. (2019). Nasr, Shokri, and Houmansadr further explored the use of scaling as a denoising strategy to optimize DP utility in Differential Privacy Stochastic Gradient Descent (DP-SGD), scaling the noisy gradients based on their usefulness Nasr et al. (2020). Subsequently, Xu et al. employed scaling and masking as post-processing denoising techniques on top of Gaussian noise-injected intermediate results in split learning, aiming to reduce the noisy neural network output's estimation error without compromising privacy Xu et al. (2022).

## 3   METHODOLOGY

### 3.1   PRELIMINARIES

#### 3.1.1   LDP

Differential privacy (DP) Dwork (2006); Dwork et al. (2014) is considered the gold standard for data privacy. Its definition is as follows:

**Definition 1 (($\epsilon, \delta$)-Differential Privacy)** *A randomized mechanism $M$ with domain $D$ and range $R$ preserves ($\epsilon, \delta$)-differential privacy if and only if for any two neighboring datasets $D, D' \in D$ and for any subset $S \subseteq R$, the following inequality holds:*

$$\Pr[M(D) \in S] \leq e^\epsilon \Pr[M(D') \in S] + \delta$$

*where $\epsilon$ is the privacy budget and $\delta$ is the failure probability.*

Local differential privacy (LDP) is a particular case of DP, where the server is not trusted and data privatization is conducted by the client. For any inputs $x$, $x' \in D$, LDP requires a randomized mechanism $M$ to satisfy:

$$\Pr[M(x) \in S] \leq e^\epsilon \Pr[M(x') \in S] + \delta \tag{1}$$

for any measurable subset subset $S \subseteq Range(M)$.

#### 3.1.2   $d_\chi$-PRIVACY

In the context of local privacy preservation, we employ $d_\chi$-privacy Chatzikokolakis et al. (2013), a specialized variant of local differential privacy tailored for textual data Feyisetan et al. (2019); Qu

et al. (2021a). $d_\chi$-privacy allows to impose high probability of observing the same output for inputs with similar semantics. We state the formal definition in the following:

**Definition 2** ($d_\chi$-**privacy**) *For an input domain $X$ and an output domain $Y$, $d_\chi$ serves as a metric space over $X$. A stochastic mechanism $M : X \to Y$ is said to adhere to $\eta d_\chi$-privacy if, for any two elements $x, x' \in X$, the output distributions $M(x)$ and $M(x')$ satisfy the following inequality:*

$$\frac{P(M(x) = y)}{P(M(x') = y)} \le e^{\eta d_\chi(x,x')}, \quad \forall y \in Y,$$

*where $\eta \ge 0$ is a tunable privacy parameter that modulates the level of privacy protection.*

The privacy guarantee indicates that the log-likelihood ratio of producing the same outcome $y$ is bounded by $\eta d_\chi(x, x')$ for any two possible inputs $x, x'$.

### 3.2 ARCHITECTURE

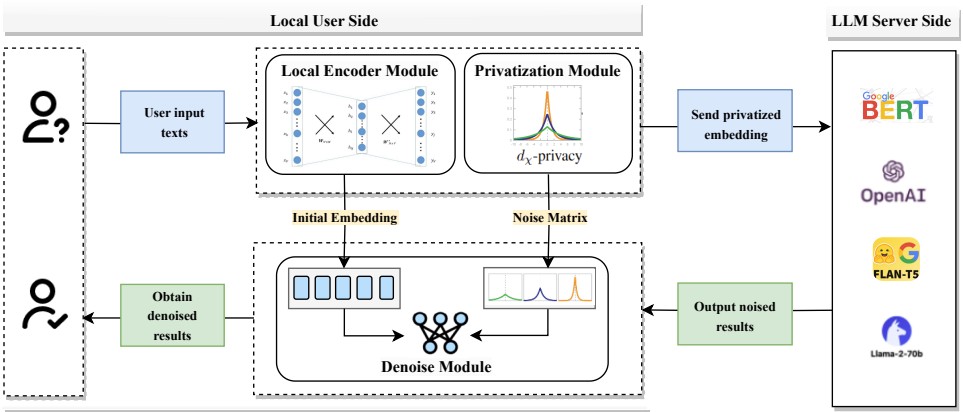

Figure 1: Overview of our privacy-preserving SnD framework.Users first obtain an initial embedding from a local encoder, followed by a noise addition via the privatization module. This privatized embedding is then transmitted to the server for processing. Upon completion, users receive a noised output, which is subsequently refined using a pre-trained denoising model to achieve an optimal balance between privacy and utility.

Denote $G : \mathcal{V}^n \to \mathbb{R}^d$ as the language model that maps $n$-token to embedding. In Split-N-Denoise (SnD), we split the language model $G$ into a local encoder $G_l : \mathcal{V}^n \to \mathbb{R}^{n \times d}$ at user side and a cloud encoder $G_c : \mathbb{R}^{n \times d} \to \mathbb{R}^d$ at server side. The local encoder consists of only the token representation layer to minimize the computation cost for user, and the server performs subsequent operations on the IRs uploaded by the clients. The architecture of SnD is depicted in figure, containing four main components:

- *Local encoder module*: the user retrieves the token embeddings of their input locally.
- *Privatization module*: the token representations are privatized by the user before being transmitted to the server to satisfy LDP.
- *Cloud encoder module*: the server performs transformation on the privatized token representations and returns the embedding to user.
- *Denoise module*: user conducts local denoising on the received embedding leveraging their raw inputs and specific noise levels.

### 3.3 NOISE MECHANISM

We adopt $d_\chi$-privacy to privatize the token representation layers on user side. Given an input sequence $x = [x_1, \ldots, x_n]$, the token representation layer transforms $x$ into a vector sequence

$X = [\boldsymbol{x}_1, \dots, \boldsymbol{x}_n] \in \mathbb{R}^{n \times d}$ via embedding model $E \in \mathbb{R}^{|\mathcal{V}| \times d}$, where $|\mathcal{V}|$ denotes the vocabulary size and $d$ represents the dimensionality of the embeddings.

Assuming $L_2$ norm as the distance metric, the application of $d_X$ privacy, parameterized by $\eta$, to a given word embedding $\boldsymbol{x}_t \in \mathbb{R}^d$ is realized by the addition of Laplacian noise $z \sim c\exp(-\eta||z||)$, where $c$ is a real-valued constant Wu et al. (2017). To sample $z$ from the Laplacian distribution, consider $z = l\boldsymbol{v}$, where $l$ is sampled from a Gamma distribution $\Gamma(d, 1/\eta)$ and $\boldsymbol{v}$ is uniformly sampled from the unit ball $B^d$. Consequently, the privatized representation $M(\boldsymbol{x}_t)$ can be succinctly expressed as:

$$M(\boldsymbol{x}_t) = \boldsymbol{x}_t + \boldsymbol{z}.$$

The supports for $z$ and thus $M(\boldsymbol{x}_t)$ are unbounded, imposing difficulties on subsequent denoise procedures, especially under low level of $\eta$. To improve the performance of denoise model introduced in Section 3.4, the client clips the $l_2$ norm of the privatized representation within $C_{x_t}$:

$$M'(\boldsymbol{x}_t) = \min\left(M(\boldsymbol{x}_t), M(\boldsymbol{x}_t) \cdot C_{x_t}/\|M(\boldsymbol{x}_t)\|\right) \tag{2}$$

, where $C_{x_t} = \max_{\boldsymbol{x}_t \in \mathcal{X}_t} \|\boldsymbol{x}_t\|$ is chosen to be the upper bound of $\boldsymbol{x}_t$. The user then updates its noise matrix locally according to the clipped representations for subsequent denoise. Appendix A.10 demonstrates the benefits of norm clipping empirically.

The following theorem states that the noise mechanism $M' : \mathbb{R}^d \to \mathbb{R}^d$ adheres to $\eta d_\chi-$privacy. Refer to appendix A.2 for the proof.

**Theorem 1** *For any $d \geq 1$ and any $\eta > 0$, the mechanism $M' : \mathbb{R}^d \to \mathbb{R}^d$ achieves $\eta d_\chi-privacy$ with respect to $d_\chi(\boldsymbol{x}, \boldsymbol{x}') = \|\boldsymbol{x} - \boldsymbol{x}'\|$.*

## 3.4 DENOISE MODEL

**Limitation of server-side denoise:** the denoising ability of a server is limited by its lack of knowledge regarding the noise levels. The server's capacity to remove noise is inherently conflicted with the level of privacy protection. Intuitively, if the server could produce an appropriate denoised output on its own, there is a higher probability that it can also reconstruct the original user input. Proposition 1 below gives the lower bound of mean square error (MSE) for server-side denoise algorithms. The proof can be found in Appendix A.3.1.

**Proposition 1** *Let $\boldsymbol{y} \in \mathcal{Y} \subseteq \mathbb{R}^k$ be the original vector without noises added, and let $\hat{\boldsymbol{y}} \in \mathbb{R}^k$ be the noisy vector obtained under $\eta d$-privacy mechanism. Denote $D_s : \mathbb{R}^k \to \mathbb{R}^k$ as the denoising algorithm run by the server. Suppose $D_s$ is unbiased and the token embeddings are bounded by $B_x$:*

$$\|\boldsymbol{x}' - \boldsymbol{x}\| \leq B_x, \forall \boldsymbol{x}', \boldsymbol{x} \tag{3}$$

*, then:*

$$\mathbb{E}[\|D_s(\hat{\boldsymbol{y}}) - \boldsymbol{y}\|/k] \geq \frac{\sum_{i=1}^{d} \mathrm{diam_i}(\mathcal{Y})^2/4\mathrm{k}}{e^{\eta B_x} - 1} \tag{4}$$

*where $\mathrm{diam_i}(\mathcal{Y}) = \sup_{\mathrm{y,y'} \in \mathcal{Y}:\mathrm{y_j=y'_j \forall j \neq i}} |\mathrm{y_i - y'_i}|$ is the diameter of $\mathcal{Y}$ in the $i$-th dimension.*

**Remark 2** *The vector $\boldsymbol{y}$ can be: (i) the token representations uploaded from users, (ii) output embeddings, or (iii) any intermediate results returned by the language model based on the token embeddings. The instantiation of $\boldsymbol{y}$ is determined by the layer at which the server runs denoising algorithm.*

To address the limitation, we propose a denoise framework where users conduct error correction on the noisy embeddings using their specific noises and raw inputs. Given the black-box nature of neural network transformation on the privatized token representations, we propose to train a transformer-based model for embedding denoise.

Let $\tilde{X} = [\tilde{\boldsymbol{x}}_1, \dots, \tilde{\boldsymbol{x}}_n]$, $Z = [\boldsymbol{z}_1, \dots, \boldsymbol{z}_n] \in \mathbb{R}^{n \times d}$ denote, respectively, the privatized token representations and noise matrix. Noted that the noise vector is updated with the clipped privatized token embeddings $\boldsymbol{z} = M'(\boldsymbol{x}_t) - \boldsymbol{x}_t$. After a series of operations, the server returns a noisy embedding

$e_n$ capturing the context of input token to the user. The denoise model is parameterized by a $L$-layer transformer decoder, $D : \mathbb{R}^{(2n+1)\times d} \to \mathbb{R}^d$:

$$e_d = D(e_n, \tilde{X}, Z) \tag{5}$$

The input to the denoise model $H_0$ is a concatenation of vectors:

$$H_0 = [e_n; \tilde{x}_1, \ldots, \tilde{x}_n; z_1, \ldots, z_n] \tag{6}$$

Let $h_t^l$ represents the hidden state for the $t^{th}$ vector at layer $l$. This state is computed using the following recursive relation:

$$h_t^l = h_t^{l-1} + a_t^{l-1} + m_t^{l-1} \tag{7}$$

where

$$a_t^{l-1} = attn^l(h_1^{l-1}, h_2^{l-1}, ..., h_{2n+1}^{l-1}), \; m_t^{l-1} = W_{proj}^l \sigma(W_{fc}^l \gamma(a_t^l + h_t^{l-1})) \tag{8}$$

The denoised embedding is obtained directly from the hidden state representation for $e_n$ at the final layer:

$$e_d = h_0^L \tag{9}$$

We visualize the architecture of the denoise model in figure 3. Intuitively, the noisy embedding undergoes $L$ steps to transform into the denoised embedding. In each step, the transformation is conditioned on the feature representations of raw IRs as well as specific noises.

To train a denoise model, the server samples a set of noises added to the token representations of public corpus. Subsequently, the clean embedding $e_c$ and noisy embedding $e_n$ are computed from, respectively, the raw and privatized token representations:

$$e_c = G(X), \; e_n = G(\tilde{X}) \tag{10}$$

The denoise model is trained on the above datasets with the objective to minimize the deviation between denoised and clean embeddings:

$$\min_D \mathbb{E}[\|D(e_n, \tilde{X}, Z) - e_c\|^2] \tag{11}$$

The pretrained model is shared with users to conduct denoising on the received embeddings locally. It is important to note that the denoise model does not expose any information regarding user data. This is primarily due to the fact that the model's training is carried out exclusively on a public dataset, rendering it irrelevant to users' private inputs.

### 3.5 COMPLEXITY ANALYSIS

In this section, we analyze the communication complexity and user computation complexity of our framework.

*Communication complexity*: the communication cost can be broken as: (1) user uploads the token representations to the server ($O(nd)$ messages); (2) server share the embeddings with user ($O(d)$ messages). Hence, the total communication overhead is $O(nd)$.

*User computation complexity*: user's computation cost can be broken as: (1) retrieving token embeddings from input text ($O(n)$ complexity); (2) performing local denoising with the transformer-based model ($O(n^2dL)$ complexity Vaswani et al. (2017)). Therefore, the user's computation cost adds up to $O(n^2dL)$.

## 4 EXPERIMENTAL RESULTS

### 4.1 EXPERIMENT SETTUP

We evaluate our framework on three classes of LLMs: Bert Devlin et al. (2018), GPT2 Radford et al. (2019), and T5 Raffel et al. (2020). The architectures of our denoise and downstream models are described in appendix A.6. We benchmark our experiments against three baseline methods: (i) Token embedding privatization (TokEmbPriv) Qu et al. (2021b), where the token embeddings are perturbed

by the user before sending them to the server. (ii) Text-to-text privatization (Text2Text) Feyisetan et al. (2019); Qu et al. (2021b), where the plain token sequence is transformed into a privatized token sequence by replacing each word with the perturbed token embeddings. (iii) Privacy-Preserving Prompt Tuning (RAPT) Li et al. (2023) that protects prompt tuning and inference with local DP.

To assess the performance of our approach, we employ two distinct evaluation metrics: (1) similarity with $e_c$: we compute the mean square error (MSE) and cosine similarity (COS) between $e_c$ and $e_d$, the clean and privatized embeddings, to quantify the extent of data variations induced by the perturbation process; (2) performance on downstream tasks: we utilize accuracy scores (ACC) and area under the roc curve (AUC) to gauge the utility of the embeddings on downstream tasks.

## 4.2 DATASETS

To train the denoise model, we use the combination of 20 datasets to better mimic the generalized training scenarios, including TweetEval Offensive Barbieri et al. (2020), Hate Speech 18 de Gibert et al. (2018), Health Fact Kotonya & Toni (2020), Daily Dialogue Li et al. (2017), etc. See the full list of datasets we used in Appendix A.4.

We test our denoising performance on a collection of downstream tasks: (i) Sentence classification: CoLA Warstadt et al. (2019), (ii) Pair similarity: Quora Question Pairs (QQP) Chen et al. (2018), MSR Paraphrase Corpus (MRPC) Dolan & Brockett (2005), (ii) Recognizing Textual Entailment (RTE) Dagan et al. (2006); Bar-Haim et al. (2006); Giampiccolo et al. (2007); Bentivogli et al. (2009). Refer to appendix A.5 for the evaluation details.

## 4.3 ATTACKS

We simulate two inference attacks on the privatized token embeddings from SnD to investigate the privacy protection ability under varying $\eta$.

**Embedding inversion attack Li et al. (2023); Qu et al. (2021b):** a token-level attack that reconstructs the raw text from the privatized token representation. Given a noisy embedding $\hat{x}_t, t \in [1, n]$, the server identify a token $x_t$ closest to $\hat{x}_t$ measured by $L_2$ distance in the embedding space:

$$x_t = \arg\min_k \|w_k - \hat{x}_t\| \tag{12}$$

, where $w_k$ represents the representation for the $k^{th}$ token in the vocabulary.

**Attribute inference attack Li et al. (2023):** an attack that infers the sensitive features of records from the privatized token representations. We rely on the twitter text dataset Vashisth & Meehan (2020) to predict the gender based on the user's review.

## 4.4 EXPERIMENT RESULTS

### 4.4.1 PERFORMANCE ON DOWNSTREAM TASK

We record the performance on various downstream task in terms of accuracy (ACC) under varing $\eta$ in Table 1, 2 and 3. The utility is benchmarked against the case without any noise injection and thus no denoise operation, denoted by $\eta = \infty$. One important observation is that our framework maintains acceptable accuracy compared with the non-privatized setting. Across the chosen $\eta$ levels and four downstream tasks, Bert, T5, and GPT models yield average model losses of 4.31%, 4.48%, and 5.25%, respectively. It is observed that larger models tend to incur greater utility loss, which aligns with the intuitive understanding that transformed noises become increasingly unpredictable—and consequently, more challenging to denoise—after traversing through additional layers. Noted that we perform evaluation on the embeddings from pre-trained model without any fine-tuning, and thus there's a gap between the accuracy in our results for $\eta = \infty$ and the SOTA benchmarks.

### 4.4.2 COMPARISON WITH BASELINE

In Table 4, 5, and 6, we assess and compare the performance of three model families against three baseline methods using AUC. For the three model families, we selected three distinct $\eta$ levels for experimentation, given the varying noise tolerance of each model. Note that $\eta$ levels do not possess

Table 1: Accuracies on downstream tasks for BERT.

| $\eta$ | DistillBert (66m) | | | Bert Base (110m) | | | Bert Large (340m) | | |
|---|---|---|---|---|---|---|---|---|---|
| | 100 | 500 | $\infty$ | 100 | 500 | $\infty$ | 100 | 500 | $\infty$ |
| CoLA | 0.693 | 0.694 | 0.701 | 0.688 | 0.694 | 0.751 | 0.697 | 0.699 | 0.757 |
| QQP | 0.632 | 0.649 | 0.683 | 0.667 | 0.688 | 0.728 | 0.676 | 0.684 | 0.706 |
| MRPC | 0.683 | 0.691 | 0.695 | 0.689 | 0.725 | 0.742 | 0.684 | 0.689 | 0.701 |
| RTE | 0.578 | 0.580 | 0.592 | 0.592 | 0.610 | 0.616 | 0.590 | 0.601 | 0.621 |

Table 2: Accuracies on downstream tasks for T5.

| $\eta$ | T5 Small (60m) | | | | T5 Base (220m) | | | | T5 Large (770m) | | | |
|---|---|---|---|---|---|---|---|---|---|---|---|---|
| | 0.001 | 0.01 | 1 | $\infty$ | 0.001 | 0.01 | 1 | $\infty$ | 0.001 | 0.01 | 1 | $\infty$ |
| CoLA | 0.69 | 0.69 | 0.69 | 0.71 | 0.69 | 0.70 | 0.70 | 0.73 | 0.70 | 0.70 | 0.70 | 0.75 |
| QQP | 0.68 | 0.69 | 0.68 | 0.71 | 0.66 | 0.67 | 0.69 | 0.72 | 0.66 | 0.67 | 0.70 | 0.71 |
| MRPC | 0.68 | 0.69 | 0.69 | 0.70 | 0.69 | 0.69 | 0.70 | 0.71 | 0.68 | 0.69 | 0.69 | 0.71 |
| RTE | 0.55 | 0.56 | 0.58 | 0.60 | 0.57 | 0.58 | 0.62 | 0.63 | 0.57 | 0.59 | 0.61 | 0.62 |

a universal implication across model families, as varying models exhibit distinct robustness against inference attacks, as delineated in Section 4.4.3.

Table 3: Accuracies on downstream tasks for GPT2.

| $\eta$ | GPT2 Small (120m) | | | GPT2 Medium (345m) | | | GPT2 large (774m) | | | GPT2 Xlarge (1.5b) | |
|---|---|---|---|---|---|---|---|---|---|---|---|
| | 1 | 100 | $\infty$ | 1 | 100 | $\infty$ | 1 | 100 | $\infty$ | 100 | $\infty$ |
| CoLA | 0.688 | 0.700 | 0.709 | 0.690 | 0.698 | 0.728 | 0.700 | 0.701 | 0.724 | 0.693 | 0.766 |
| QQP | 0.645 | 0.657 | 0.716 | 0.647 | 0.652 | 0.711 | 0.637 | 0.650 | 0.721 | 0.650 | 0.741 |
| MRPC | 0.688 | 0.691 | 0.720 | 0.688 | 0.693 | 0.710 | 0.674 | 0.691 | 0.701 | 0.686 | 0.705 |
| RTE | 0.556 | 0.563 | 0.581 | 0.567 | 0.578 | 0.583 | 0.581 | 0.606 | 0.611 | 0.584 | 0.592 |

Table 4: AUC comparisons for BERT models with QQP task.

| $\eta$ | DistillBert | | | Bert Base | | | Bert Large | | |
|---|---|---|---|---|---|---|---|---|---|
| | 50 | 100 | 500 | 50 | 100 | 500 | 50 | 100 | 500 |
| TokenEmbPriv | 0.502 | 0.518 | 0.521 | 0.511 | 0.535 | 0.557 | 0.522 | 0.525 | 0.541 |
| Text2Text | 0.541 | 0.541 | 0.541 | 0.512 | 0.513 | 0.513 | 0.507 | 0.537 | 0.540 |
| RAPT | 0.517 | 0.515 | 0.545 | 0.513 | 0.528 | 0.551 | 0.515 | 0.539 | 0.565 |
| **SnD** | **0.583** | **0.600** | **0.610** | **0.674** | **0.675** | **0.691** | **0.639** | **0.655** | **0.657** |

Table 5: AUC Comparison for GPT Models with MRPC task.

| $\eta$ | GPT2 Small | | | GPT2 Medium | | | GPT2 large | | |
|---|---|---|---|---|---|---|---|---|---|
| | 1 | 50 | 100 | 1 | 50 | 100 | 1 | 50 | 100 |
| TokenEmbPriv | 0.514 | 0.525 | 0.532 | 0.526 | 0.523 | 0.530 | 0.512 | 0.513 | 0.518 |
| Text2Text | 0.498 | 0.502 | 0.502 | 0.496 | 0.498 | 0.498 | 0.491 | 0.499 | 0.500 |
| RAPT | 0.504 | 0.521 | 0.524 | 0.503 | 0.502 | 0.539 | 0.500 | 0.510 | 0.547 |
| **SnD** | **0.542** | **0.552** | **0.579** | **0.553** | **0.578** | **0.573** | **0.547** | **0.556** | **0.556** |

Table 6: AUC Comparison for T5 Models with RTE task.

| $\eta$ | T5 Small | | | T5 Base | | | T5 Large | | |
|---|---|---|---|---|---|---|---|---|---|
| | 0.001 | 0.01 | 0.1 | 0.001 | 0.01 | 0.1 | 0.001 | 0.01 | 0.1 |
| TokenEmbPriv | 0.503 | 0.515 | 0.514 | 0.505 | 0.525 | 0.537 | 0.518 | 0.503 | 0.537 |
| Text2Text | 0.512 | 0.533 | 0.537 | 0.504 | 0.527 | 0.537 | 0.501 | 0.507 | 0.516 |
| RAPT | 0.510 | 0.548 | 0.547 | 0.506 | 0.532 | 0.533 | 0.514 | 0.519 | 0.516 |
| **SnD** | **0.547** | **0.577** | **0.575** | **0.566** | **0.564** | **0.611** | **0.566** | **0.580** | **0.599** |

For each model family, a representative task was selected. For BERT models, SnD outperforms TokenEmbPriv, Text2Text, and RAPT by an average of 22.2%, 22.1%, and 20.9%, respectively. For GPT models, SnD results in AUC higher than the three baselines from 7.3% to 12.3% on average. For T5 models, the performance of SnD is higher than the baselines by an average of over 10%. It can be observed that TokenEmbPriv and Text2Text exhibit poorer performance compared to the other two approaches. This could be attributed to the lack of denoise or reconstruction mechanism within these methods. Furthermore, the unbounded noise support in TokenEmbPriv leads to significant deviations between the privatized token representations and their original values. The MSE and COS between the initial and recovered embeddings in presented in Appendix A.8. Both AUC and the similarity metrics suggest our technique's proficiency in restoring the original attributes of the noised embedding after procuring the perturbed results from the server.

### 4.4.3 INFERENCE ATTACK

In this section we present the results for embedding inversion attack, and the discussion for attribute inference attack can be found in Appendix A.7. Figure 2 visualizes the attack accuracy, measured by the percentage of token correctly identified by the attack, for the three series of models at various $\eta$ values. For Bert models, the attack success rates remain below $1\%$ with $\eta \leq 500$. GPT models exhibit negligible attack accuracy with $\eta$ values up to 100, while GPT Xlarge demonstrates exceptional robustness against inference attacks as $\eta$ increases. T5 models, on the other hand, require much smaller privacy budgets to resist inference attacks effectively.

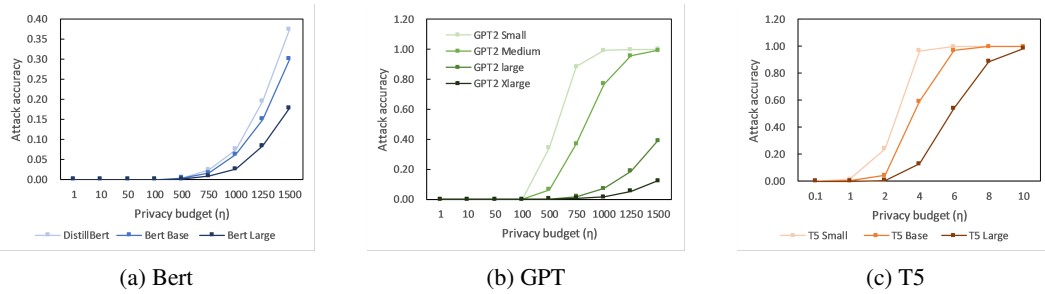

| (a) Bert | (b) GPT | (c) T5 |
|---|---|---|

Figure 2: Embedding inversion attack accuracy under varying $\eta$.

## 5 CONCLUSION

This paper proposes SnD, a framework that employs split inference and denoising techniques to protect LLM inference with LDP. We split the language model to deploy the token representation layer on user side. User perturbs the token embeddings to guarantee $d_\chi$-privacy before transmitting them to the server. To improve the utility of embeddings, user conducts local denoising with a pre-trained model leveraging the raw token representations and specific noises. The empirical studies show that SnD performs better in maintaining the utility of embeddings compared with baseline methods by over 10% on average. Our study opens up new possibilities for privacy-preserving LLM inference, in terms of scalability to larger LLM, optimizing user computation cost, and extension to sequence-to-sequence inference model (see Appendix A.12).

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

## A  APPENDIX

### A.1  COMPARISON OF PRIVACY-PRESERVING LLM INFERENCE

Table 7 summarizes the existing privacy-preserving LLM inference approaches along four dimensions: (i) involvement of finetuning, including parameter efficient finetunings, on task specific data, (ii) adoption of server-side denoise technique on privatized values, (iii) adoption of user-side denoise technique on privatized values, (iv) privacy guarantee in terms of the security in multiparty computation (SMPC), or local differential privacy (LDP). Noted that RAPT employs a reconstruction head to improve the robustness of prompt tuning process, where the module reconstructs the random tokens to help the LLM better understand the privatized token at training stage. However, the precise mechanism by which the LLM learns to decode the privatized tokens remains unclear, especially considering that the reconstruction module works solely on random tokens. Furthermore, the reconstruction head is discarded during LLM inference stage, rendering no denoise mechanism for LLM inference.

Table 7: Comparison of different privacy-preserving LLM inference approaches.

|  | PCFT | TokEmbPriv | Text2Text | RAPT | SnD |
|---|---|---|---|---|---|
| Finetuning | × | × | × | √ | × |
| Server denoise | × | × | × | √ | × |
| User denoise | × | × | × | × | √ |
| Privacy guarantee | SMPC | LDP | LDP | LDP | LDP |

### A.2  PROOF OF THEOREM 1

To prove the theorem, we first demonstrate that the noise follows Laplacian distribution:

**Lemma 1** *By sampling $l$ from Gamma distribution $\Gamma(d, 1/\eta)$, and $\boldsymbol{v}$ from the unit ball $B^d$, the vector $z = l\boldsymbol{v}$ can be released as $d$-dimensional Laplacian $z \sim c\exp(-\eta\|z\|)$.*

**Proof 1** *The proof follows by changing variables to spherical coordinates and showing that the Laplacian can be expressed as the product of $\boldsymbol{v}$ and $l$. See, for instance, Lemma 4 in Fernandes et al. (2019).*

We can now proceed to the proof of Theorem 1.

**Proof 2** *Plugging in the probability density function of $z$, it holds that:*

$$\frac{P(M(\boldsymbol{x}) = \boldsymbol{y})}{P(M(\boldsymbol{x}') = \boldsymbol{y})} = \frac{P(z = \boldsymbol{y} - \boldsymbol{x})}{P(z = \boldsymbol{y} - \boldsymbol{x}')} = \exp(\eta(\|\boldsymbol{y} - \boldsymbol{x}'\| - \|\boldsymbol{y} - \boldsymbol{x}\|)) \le \exp(\eta(\|\boldsymbol{x}' - \boldsymbol{x}\|)) \quad (13)$$

*, for any $\eta > 0$.*

*Then the mechanism $M'$ achieves $\eta d_\chi$-DP based on post-processing property.*

## A.3 DENOISE MODEL

### A.3.1 PROOF OF PROPOSITION 1

We begin with the below Lemma to bound the relative entropy of $\hat{\boldsymbol{y}}$ and $\hat{\boldsymbol{y}}'$.

**Lemma 2** *Let $\hat{\boldsymbol{y}} \in \mathbb{R}^k$ be the noisy vector described in Proposition 1. Denote $F : \mathbb{R}^{n \times d} \to \mathbb{R}^k$ as the transformation from privatized token representations to $\hat{\boldsymbol{y}}$. It holds that:*

$$\mathbb{D}_2(F(\hat{\boldsymbol{y}})\|F(\hat{\boldsymbol{y}}')) \le \mathbb{D}_\infty(F(\hat{\boldsymbol{y}})\|F(\hat{\boldsymbol{y}}'))) \le \eta B_x, \forall \hat{\boldsymbol{y}} = M'(\boldsymbol{x}), \hat{\boldsymbol{y}}' = M'(\boldsymbol{x}') \quad (14)$$

*where $\mathbb{D}_i(\cdot)$ denotes the rényi divergence of order $i$.*

**Proof 3** *The proof directly follows by applying post-processing properties on the relative entropy.*

Then we proceed to the proof of Proposition 1. Let $h = D_s(\hat{\boldsymbol{y}})$ For an unbiased denoise model, the MSE is lower bounded by:

$$\mathbb{E}[\|D_s(\hat{\boldsymbol{y}}) - \boldsymbol{y}\|/k] \ge \sum_i Var\left(D_s(\hat{\boldsymbol{y}})_i\right) \quad (15)$$

Then we examine the bound of $Var\left(D_s(\hat{\boldsymbol{y}})_i\right)$. Denote $h = D_s(\hat{\boldsymbol{y}})$ as the denoised output and $\mu(\boldsymbol{y})$ as the expectation of $h$. From Hammersley-Chapman-Robbins Bound, we have:

$$
\begin{aligned}
Var\left(D_s(\hat{\boldsymbol{y}})_i\right) &\ge \frac{(\mu(\boldsymbol{y} + e_i)_i - \mu(\boldsymbol{y})_i)^2}{\mathbb{E}[(p(h; \boldsymbol{y} + e_i)/p(h; \boldsymbol{y}) - 1)^2]} \ge \frac{(\mu(\boldsymbol{y} + e_i)_i - \mu(\boldsymbol{y})_i)^2}{e^{\eta B_x} - 1} \\
&\overset{(a)}{\ge} \frac{\sum_{i=1}^d \mathrm{diam_i}(\mathcal{Y})^2/4\mathrm{k}}{e^{\eta B_x} - 1}
\end{aligned} \quad (16)
$$

where $\mathbb{E}[\cdot]$ is the expectation taken over $p(h; \boldsymbol{y})$, $p(h; \boldsymbol{y})$ is the density function of $h$ given $\boldsymbol{y}$, and $e_i$ is the standard basis vector with ith coordinate equal to 1. (a) follows from the unbias property of the denoise model (see, for example, Theorem A.1 in Guo et al. (2022)).

### A.3.2 FIGURE OF DENOISE ARCHITECTURE

## A.4 DETAILS OF DATASETS

### A.4.1 DATASET TO TRAIN DENOISE MODEL

**SQuAD**: The Stanford Question Answering Dataset (SQuAD) is a reading comprehension dataset, with questions posed by crowdworkers based on a set of Wikipedia articles. The answer to every question is a segment of text from the corresponding article, or the question might be unanswerable Rajpurkar et al. (2016).

**AG News**: This dataset contains more than 1 million news articles, categorizing text into classes like sports, business, and tech Zhang et al. (2015).

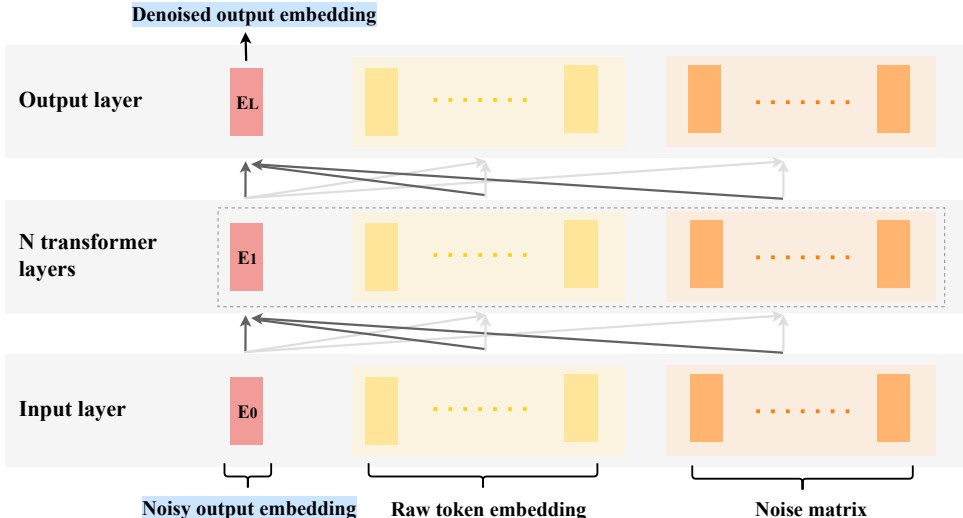

Figure 3: Architecture of denoise model.The denoise model accepts the noised output embedding from the LLM model, in conjunction with the raw token embedding and noise matrix, as input. Through multiple transformers, the model learns to denoise, ultimately producing a denoised output embedding to augment the performance of downstream tasks.

**Financial Phrasebank**: Comprising sentiments in the financial domain, specifically with sentences where all annotators concur. It is primarily for 3-class sentiment analysis Malo et al. (2014).

**Banking77**: It contains online banking queries annotated with their corresponding intents, focusing on fine-grained single-domain intent detection Casanueva et al. (2020).

**Health Fact**: A comprehensive dataset for explainable automated fact-checking of public health claims Kotonya & Toni (2020).

**Poem Sentiment**: A dataset for 4-class sentiment analysis on poem verses from Project Gutenberg Sheng & Uthus (2020).

**Tweet Eval - Sentiment**: Containing tweets for sentiment analysis Barbieri et al. (2020).

**Tweet Eval - Emotion**: Comprising tweets labeled with specific emotions Barbieri et al. (2020).

**Tweet Eval - Hate**: A dataset to classify tweets containing hate speech Barbieri et al. (2020).

**Tweet Eval - Offensive**: A dataset for classifying tweets deemed offensive Barbieri et al. (2020).

**ADE Corpus V2**: A dataset for classification if a sentence discusses Adverse Drug Reaction or not. This dataset also extracts the relation between Adverse Drug Event and Drug Gurulingappa et al. (2012).

**Hate Speech18**: Dedicated to detecting hate speech in texts extracted from Stormfront, a white supremacist forum de Gibert et al. (2018).

**SMS Spam**: Comprising SMS labeled texts, this dataset is utilized to identify spam messages Almeida et al. (2011).

**Daily Dialog**: A high-quality multi-turn dialog dataset contains dialogues derived from daily conversations Li et al. (2017).

**Yelp Review Full**: Comprising reviews from Yelp for text classification. This dataset is extracted from the Yelp Dataset Challenge 2015 data Zhang et al. (2015).

**App Reviews**: A dataset of user reviews of Android applications belonging to different categories Grano et al. (2017).

**Amazon Polarity**: Contains reviews from Amazon, including product and user information, ratings, and a text review McAuley & Leskovec (2013); Zhang et al. (2015).

**Rotten Tomatoes**: A movie review dataset used for sentiment analysis. This dataset comprises reviews from the Rotten Tomatoes website (Pang & Lee, 2005).

**Wikitext**: A collection of over 100 million tokens extracted from the set of verified Good and Featured articles on Wikipedia. Used for language modeling and other NLP tasks (Merity et al., 2016).

**OpenWebText**: An open-source collection of web articles, modeled after the dataset used in the original "GPT" work (Gokaslan* et al., 2019).

### A.4.2 DATASET FOR DOWNSTREAM TASKS

**QQP**: The Quora Question Pairs2 dataset consists of question pairs to determine semantic equivalence Chen et al. (2018).

**RTE**: The Recognizing Textual Entailment (RTE) datasets aggregates multiple Recognizing Textual Entailment challenges, determining if texts entail each other. It combines data from several RTE challenges Dagan et al. (2006); Bar-Haim et al. (2006); Giampiccolo et al. (2007); Bentivogli et al. (2009).

**MPRC**: The Microsoft Research Paraphrase Corpus (MRPC) contains sentence pairs extracted from online news sources, with labels to predict the equivalence of the sentences in the pair Dolan & Brockett (2005).

**CoLA**: The Corpus of Linguistic Acceptability (CoLA) consists of sentences from books and journal articles on linguistic theory with annotations for acceptability (grammaticality) Warstadt et al. (2019).

### A.5 EVALUATION OF DOWNSTREAM TASKS

We follow the steps below to conduct evaluation on downstream tasks:

- Obtain the embeddings of text in training and testing datasets via privacy-preserving LLM inference framework.
- Train a classification model on the privatized (denoised) embeddings from training set.
- Test the performance of classification model on the privatized (denoised) embeddings from testing set.

### A.6 SPECIFICATIONS ON EXPERIMENTAL SETTING

#### A.6.1 EXPERIMENTAL SETTINGS

All the experiments are performed on a virtual server with Intel Xeon Platinum 8336C CPU and NVIDIA RTX A6000 GPU (CUDA version 12.2). We utilize Python 3.10 as the programming language and pytorch 1.13 as the underlying framework.

#### A.6.2 HYPERPARAMETERS OF DENOISE MODEL

The hyperparameters of denoise model are represented as followed:

- $d_{model}$: Dimension of input embeddings and hidden states.
- $d_{ff}$: Hidden dimension in the feed forward network.
- $d_{kv}$: Dimension of each head in the multi-head attention layer.
- $n_{head}$: Number of heads in the multi-head attention layer.
- $L$: Number of layers.

Table 8 lists the hyperparameters for each denoise model.

Table 8: Hyperparameters of denoise models.

|              | $d_{model}$ | $d_{ff}$ | $d_{kv}$ | $n_{head}$ | $L$ |
|--------------|-------------|----------|----------|------------|-----|
| DistillBert  | 768         | 768      | 240      | 6          | 3   |
| Bert Base    | 768         | 1024     | 240      | 8          | 6   |
| Bert Large   | 1024        | 1024     | 256      | 8          | 6   |
| T5 Small     | 512         | 512      | 240      | 6          | 6   |
| T5 Base      | 768         | 768      | 256      | 8          | 6   |
| T5 Large     | 1024        | 1024     | 256      | 8          | 6   |
| GPT2 Small   | 768         | 768      | 240      | 8          | 6   |
| GPT2 Medium  | 1024        | 1024     | 256      | 8          | 6   |
| GPT2 Large   | 1280        | 1280     | 256      | 8          | 6   |
| GPT2 XLarge  | 1600        | 1600     | 256      | 10         | 6   |

### A.6.3 TRAINING OF DENOISE MODEL

We take the following measures to train the denoise model adapting to varying $\eta$ levels:

- Divide the privacy budget $\eta$ into three groups. For each model, we partition $\eta$ into three groups according to the correlation coefficient between the plain and privatized token embeddings.

- Train separate denoise models for each group of $\eta$. For each partition, we sample the noises from two representative $\eta$ levels as training inputs to the denoise model.

- Perform denoising using the denoise model corresponding to the appropriate group. During the inference stage, users specify the desired levels $\eta$ and retrieve the denoise model from the corresponding partition.

### A.6.4 TRAINING OF DOWNSTREAM CLASSIFICATION MODEL

For downstream classification task, we employ a simple neural network model composed of two fully connected layers and two rectified linear unit (ReLU) activation functions. We set the hidden dimension to be the same as input dimension. Regarding pairwise similarity and textual entailment tasks, we concatenate the embeddings of both sentences into one vector which is then passed as input to the classification model.

### A.6.5 SPECIFICATION OF BASELINE

Below we list the experimental specifications of the three baseline methods. All approaches utilize the $d_\chi$-privacy definition with $L_2$ distances $d(x, x') = \|x - x'\|$. The maximum sequence length is set to 512 in SnD and baselines.

- TokEmbPriv: the user perturbed the token embedding with $d_\chi$-privacy before uploading to the server. The privatization is performed by adding random noise $Z$ sampled from a $d$-dimensional distribution with density function $p(Z) \sim \exp(-\eta\|Z\|)$.

- Text2Text: the user transforms the plain token sequence into a privatized token sequence. The tokens is first mapped to a embedding space using the embedding model given by the token representation layer $E : \mathcal{V} \to \mathbb{R}^d$, where $\mathcal{V}$ denotes set of vocabulary. The token embeddings are privatized by adding noises drawn from the exponential distribution described in TokEmbPriv. Finally, we identify the token with representation closest to the perturbed embeddings measured by Euclidean distance.

- RAPT: We adopt the prompt tuning method with prompt length set to 150. We finetune the model for 4 epochs and set the batch size to 15. The vocabulary size of the reconstruct head and plain token size are set to 7,630 and 40, respectively. We employ AdamW Loshchilov & Hutter (2018) as the optimizer and set the learning rate to 6e-5.

### A.7 ATTRIBUTE INFERENCE ATTACK

Figure 4 presents the accuracies of inference attack on the tweet review dataset. In the case of Bert models, the attack accuracies are around 0.5 when $\eta \leq 1500$. As for GPT2 small, the attack performance gradually increases as $\eta$ reaches 500, whereas for GPT2 medium and large, the attack accuracies remain consistently low when $\eta \leq 1500$. For T5 models, the attacker's inference capability starts to grow for $\eta \geq 600$.

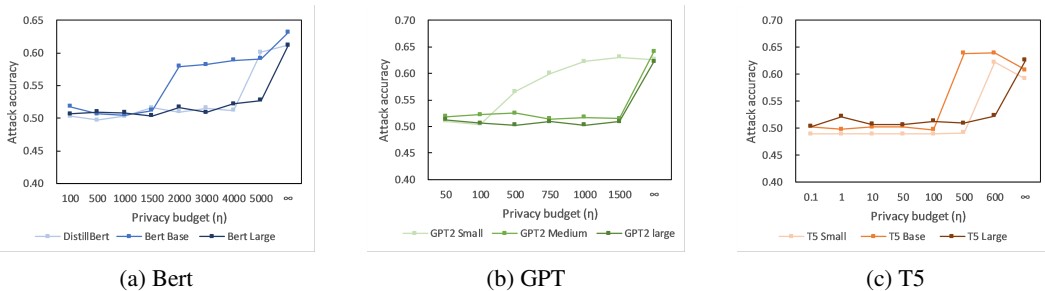

|  (a) Bert | (b) GPT | (c) T5 |

Figure 4: Attribute inference attack accuracy under varying $\eta$.

### A.8 COMPARISON WITH BASELINE IN TERMS OF SIMILARITY

Table 9, 10, and 11 compare the MSE and COS of SnD with the three baselines. A higher COS (or lower MSE) suggests that the final output embeddings is more similar to the clean output embeddings, indicating a higher preservation of utility. Noted that we don't list the metrics for RAPT as it returns the classification result to the user, which doesn't involve the transmission of output embeddings. One important observation is that our proposed methods results in significantly lower MSE and higher COS compared with the two baselines.

Table 9: MSE and COS comparisons for BERT models with QQP task.

|  |  | DistillBert | | | Bert Base | | | Bert Large | | |
|---|---|---|---|---|---|---|---|---|---|---|
| $\eta$ |  | 50 | 100 | 500 | 50 | 100 | 500 | 50 | 100 | 500 |
| TokenEmbPriv | MSE | 0.507 | 0.507 | 0.630 | 0.477 | 0.475 | 0.461 | 0.629 | 0.632 | 0.630 |
|  | COS | 0.212 | 0.214 | 0.204 | 0.097 | 0.098 | 0.105 | 0.203 | 0.200 | 0.204 |
| Text2Text | MSE | 0.445 | 0.445 | 0.445 | 0.248 | 0.248 | 0.248 | 0.609 | 0.608 | 0.613 |
|  | COS | 0.279 | 0.279 | 0.279 | 0.470 | 0.470 | 0.470 | 0.224 | 0.226 | 0.224 |
| **SnD** | **MSE** | **0.241** | **0.260** | **0.098** | **0.060** | **0.075** | **0.035** | **0.119** | **0.139** | **0.098** |
|  | **COS** | **0.511** | **0.490** | **0.846** | **0.870** | **0.862** | **0.935** | **0.806** | **0.769** | **0.846** |

Table 10: MSE and COS comparisons for GPT models with MRPC task.

|  |  | GPT2 Small | | | GPT2 Medium | | | GPT2 large | | |
|---|---|---|---|---|---|---|---|---|---|---|
| $\eta$ |  | 1 | 50 | 100 | 1 | 50 | 100 | 1 | 50 | 100 |
| TokenEmbPriv | MSE | 97.019 | 21.962 | 18.520 | 35.680 | 32.189 | 31.463 | 2.584 | 1.920 | 1.608 |
|  | COS | 0.353 | 0.947 | 0.954 | 0.370 | 0.646 | 0.656 | 0.017 | 0.096 | 0.102 |
| Text2Text | MSE | 18.791 | 17.824 | 17.824 | 28.613 | 28.440 | 28.440 | 1.489 | 1.437 | 1.247 |
|  | COS | 0.951 | 0.954 | 0.954 | 0.613 | 0.628 | 0.628 | 0.093 | 0.107 | 0.134 |
| **SnD** | **MSE** | **4.667** | **4.611** | **4.177** | **11.721** | **10.333** | **11.951** | **0.502** | **0.501** | **0.484** |
|  | **COS** | **0.971** | **0.985** | **0.989** | **0.838** | **0.870** | **0.890** | **0.630** | **0.609** | **0.611** |

Table 11: MSE and COS comparisons for T5 models with MRPC task.

|  |  | T5 Small | | | T5 Base | | | T5 large | | |
| --- | --- | --- | --- | --- | --- | --- | --- | --- | --- | --- |
| $\eta$ |  | 0.001 | 0.01 | 0.1 | 0.001 | 0.01 | 0.1 | 0.001 | 0.01 | 0.1 |
| TokenEmbPriv | MSE | 0.630 | 0.234 | 0.201 | 0.230 | 0.131 | 0.093 | 0.212 | 0.120 | 0.098 |
|  | COS | 0.909 | 0.923 | 0.962 | 0.873 | 0.902 | 0.973 | 0.697 | 0.934 | 0.957 |
| Text2Text | MSE | 0.086 | 0.084 | 0.089 | 0.135 | 0.133 | 0.134 | 0.072 | 0.073 | 0.070 |
|  | COS | 0.923 | 0.924 | 0.923 | 0.826 | 0.825 | 0.826 | 0.834 | 0.837 | 0.837 |
| **SnD** | **MSE** | **0.035** | **0.038** | **0.007** | **0.004** | **0.006** | **0.003** | **0.022** | **0.021** | **0.005** |
|  | **COS** | **0.988** | **0.992** | **0.997** | **0.991** | **0.996** | **0.992** | **0.961** | **0.966** | **0.978** |

## A.9 OVERHEAD ANALYSIS

In this section, we evaluate the overhead on a virtual server with Intel Xeon Platinum 8336C CPU and NVIDIA RTX A6000 GPU (CUDA version 12.2).

To verify the practicability of SnD, we benchmark our framework with two encryption-based methods, Privacy-Computing Friendly Transformers (PCFT) Liu & Liu (2023) and Iron Hao et al. (2022). Table 12 presents the computation cost for the inference of one sample, where the token length is set as 128. The communication cost in SnD doesn't involve downloading of denoise model as this is a one-time cost. The state-of-art encryption approaches incurred significant overhead in terms of communication cost resulted from their multiparty computation protocols. The SnD results in more than $5000\times$ speedup for PCFT and $25000\times$ speedup for Iron.

Table 12: Overhead of SnD and encryption-based methods. *Comp.* and *Comm.* represent the computation and communication cost respectively. The computation costs are measured in seconds.

|  | SnD | | PCFT | | Iron | |
| --- | --- | --- | --- | --- | --- | --- |
|  | Comp. | Comm. (MB) | Comp. | Comm. (GB) | Comp. | Comm. (GB) |
| DistillBert | 0.026 | 0.00014 | 137.16 | 13.68 | 693.18 | 76.56 |
| Bert Base | 0.031 | 0.00014 | 420.12 | 5.7 | 2021.16 | 27.06 |

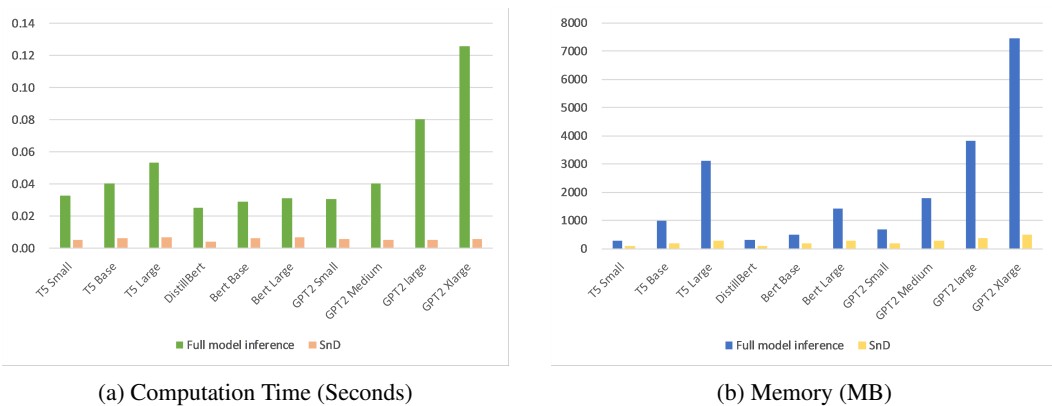

(a) Computation Time (Seconds)      (b) Memory (MB)

Figure 5: Computation and memory cost on user side. *Full model inference* denotes the case where user runs the whole language model, and *SnD* denotes the user's computation cost in our proposed method.

In Figure 5 we compare the computation and memory cost of user-side inference in two cases: (a) user only conducts initial representation retrieval and local denoising in SnD, and (b) user performs local inference with the whole language model. It can be observed that SnD has significant

advantages in terms of the overhead, and the computation benefits are greater for language models of large sizes. In particular, SnD saves the user's computation and memory cost by 95.3% and 93.5%, respectively, compared with full model inference for GPT2-Xlarge.

## A.10 ABLATION STUDIES

In this section, we conduct ablation studies to investigate the impact of user-side denoise model and norm clipping on privatized token embeddings.

**Impact of server-side denoise model:** to evaluate the impact of noise level awareness on denoising performance, we deploy the denoise model on server side using only privatized token representations as input. We train a transformer-based denoise model that outputs the denoised token representations with the objective to minimize MSE. The denoise model adopts the same hyperparameters specified in Appendix A.6. Figure 6 demonstrates that SnD significantly outperforms the server-side denoise method by over 10% in almost all cases. It's also important to note that most AUCs of server-side denoise model fall below 0.55, suggesting the server's incapacity to deduce private information about the user.

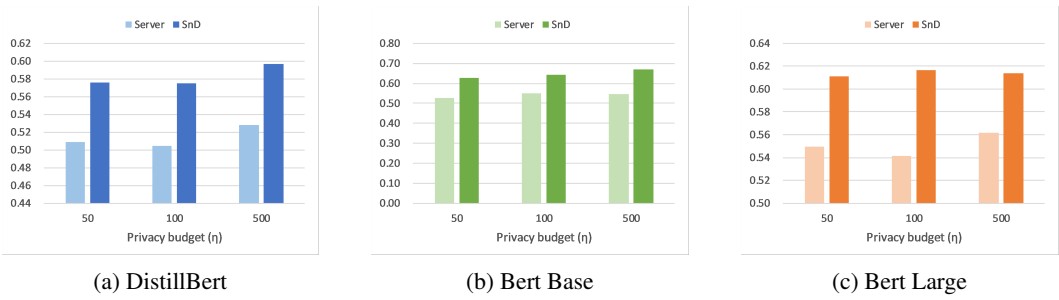

(a) DistillBert      (b) Bert Base      (c) Bert Large

Figure 6: AUC comparison on denoise model deployment for BERT models with MRPC task. *Server* denotes that the denoise model is implemented at the server side without the knowledge of noise levels.

**Impact of norm clipping:** we perform ablation studies on the norm clipping procedure for the privatized token embeddings. Figure 7 shows the AUC comparisons for three downstream tasks on T5 large model. It can be observed that clipping the privatized inputs improve the accuracy by an average of 7.5%, 15.5%, 13.4% for RTE, MRPC, and QQP tasks respectively.

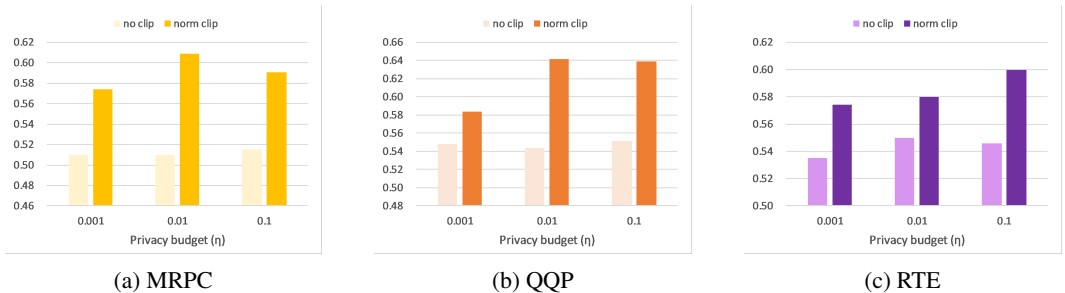

(a) MRPC      (b) QQP      (c) RTE

Figure 7: AUC on T5 Large with and without norm clipping. *No clip* refers to the case where norm clipping is not performed on the token representations.

## A.11 EXTREME LEVELS OF $\eta$

In this section, we show that our denoise mechanism allows the model to maintain the performance even under extremely low levels of privacy budget $\eta$. Table 13 presents the AUC for Bert base with $\eta$ set as 0.001, 0.01, 0.1. The correlation coefficients between the privated and clean token representations are below 0.005, indicating that the transmitted intermediate values reveal little information

about the input text. It can be observed that SnD still outperforms Text2Text by large under the low privacy settings.

Table 13: AUC for Bert Base with $\eta$ from 0.001 to 0.1.

|  | MRPC | | | | RTE | | QQP | | |
|---|---|---|---|---|---|---|---|---|---|
| $\eta$ | 0.001 | 0.01 | 0.1 | 50.001 | 0.01 | 0.1 | 0.001 | 0.01 | 0.1 |
| Text2Text | 0.525 | 0.528 | 0.520 | 0.519 | 0.520 | 0.526 | 0.510 | 0.516 | 0.527 |
| **SnD** | **0.617** | **0.616** | **0.619** | **0.576** | **0.578** | **0.569** | **0.639** | **0.650** | **0.647** |

### A.12  DISCUSSION AND FUTURE WORK

**Scalability to larger language model**: our experiments primarily focused on language models ranging from 100MB to 1GB in size. We also tested our approach on larger language model, such as LLaMa and OPT-6.7B. While we observed substantial improvements in terms of MSE and COS of the embeddings compared to the baseline, we discovered that the accuracy on downstream tasks still requires further enhancement. We suspect that the inputs undergo significantly more intricate transformations in these larger language models, necessitating the use of more sophisticated noise and denoising mechanisms.

**Reduce user computation cost**: local denoising constitutes a major component of user's computation overhead. We observe that the size of denoise model, and thus the user computation cost, scale with the underlying LLM. For those users with limited computation resource, it's crucial to design a lightweight denoise mechanism with minimal computation cost.

**Sequence-to-sequence (S2S) inference**: it's of great interest to extend our EaaS framework to S2S inference model. One important obstacle of the generalization is the noise amplification issue with S2S model. In particular, S2S relies on the auto-regressive mechanism, where the prediction of previous token is taken as an input to the next token. Therefore, the error from the previous prediction would exaggerate the deviation of the following tokens. A universal denoise model might be insufficient to correct the errors in the generated sequence.

