Our main contributions involve the following:

- To the best of our knowledge, this paper represents the pioneering effort in protect user's privacy during LLM inference with strong privacy guarantee. Existing research focuses on the privacy-preserving pre-training and fine-tuning for LLM, while few studies pay attention to the privacy concerns at inference stage, especially on privatizing user's input to guarantee DP. Our research introduces a framework to guarantee user's privacy with LDP while maintaining acceptable utility of the output.

- We propose SnD, a framework that integrates split inference and denoising techniques to protect user's privacy with LDP. We conduct empirical analysis to demonstrate the utility of embeddings on various downstream tasks.

- We design a novel denoising method deployed on user side. In this approach, a denoise model is pre-trained on server side using public dataset and synthetic noises. Subsequently, this trained model is deployed on the user side, where it leverages the specific noise levels and raw IRs provided by the user to enhance the embeddings.

## 2 PRIOR WORKS

**Local Privacy for Frozen LLMs**   With the advent of LLMs, privacy leakage has emerged as a crucial concern. Existing literature predominantly focuses on privacy conservation throughout the entire training process, encompassing pre-training Hoory et al. (2021), fine-tuning Huang et al. (2020); Kerrigan et al. (2020); Yu et al. (2021); Lukas et al. (2023), and prompt-tuning phases Duan et al. (2023); Li et al. (2023). Yet, there is a notable dearth of research addressing local privacy during the inference phase with a fully frozen LLM. This scenario, which prohibits alterations to the model's structure and parameters, is particularly complex. Nonetheless, it holds significance in black-box API access contexts, especially for proprietary models like GPT-4. One intuitive approach involves anonymizing sensitive terms prior to LLM input and subsequently restoring them post-output Kan et al. (2023); Chen et al. (2023). However, this method, while effective for obfuscating specific entities, falls short in concealing other linguistic elements, including verbs and non-named entities. Such a limitation compromises full privacy and is unsuitable for tasks necessitating exact semantic interpretation of the altered entities, such as knowledge retrieval and text continuation Chen et al. (2023). An alternative strategy might entail input text perturbation via text-to-text privatization or synthetic data generation, preserving high-dimensional features while altering human-perceivable sequences Li et al. (2023). Specifically, the text-to-text privatization projects text into a high-dimensional vector space with a pre-determined word embedding model,adding carefully calibrated noise to the vector representation, and then reconvert it to obtain the perturbed text Feyisetan et al. (2019); Qu et al. (2021a). Yet, the mere application of this technique during inference does not guarantee a satisfactory balance between privacy and

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

Refer to appendix A.1 for the proof.

## 3.4 Denoise Model

On receiving the noisy embeddings, users conduct error correction on the embeddings using their specific noises and raw inputs. Given the black-box nature of neural network transformation on the privatized token representations, we propose to train a transformer-based model for embedding denoise.

Let $\tilde{X} = [\tilde{\boldsymbol{x}}_1, \ldots, \tilde{\boldsymbol{x}}_n]$, $Z = [\boldsymbol{z}_1, \ldots, \boldsymbol{z}_n] \in \mathbb{R}^{n \times d}$ denote, respectively, the privatized token representations and noise matrix. After a series of operations, the server returns a noisy embedding $\boldsymbol{e}_n$ capturing the context of input token to the user. The denoise model is parameterized by a $L$-layer transformer decoder, $D : \mathbb{R}^{(2n+1) \times d} \to \mathbb{R}^d$:

$$\boldsymbol{e}_d = D(\boldsymbol{e}_n, \tilde{X}, Z) \tag{2}$$

The input to the denoise model $H_0$ is a concatenation of vectors:

$$H_0 = [\boldsymbol{e}_n; \tilde{\boldsymbol{x}}_1, \ldots, \tilde{\boldsymbol{x}}_n; \boldsymbol{z}_1, \ldots, \boldsymbol{z}_n] \tag{3}$$

Let $\boldsymbol{h}_t^l$ represents the hidden state for the $t^{th}$ vector at layer $l$. This state is computed using the following recursive relation:

$$\boldsymbol{h}_t^l = \boldsymbol{h}_t^{l-1} + \boldsymbol{a}_t^{l-1} + \boldsymbol{m}_t^{l-1} \tag{4}$$

where

$$\boldsymbol{a}_t^{l-1} = attn^l(\boldsymbol{h}_1^{l-1}, \boldsymbol{h}_2^{l-1}, ..., \boldsymbol{h}_{2n+1}^{l-1}), \ \boldsymbol{m}_t^{l-1} = W_{proj}^l \sigma(W_{fc}^l \gamma(\boldsymbol{a}_t^l + \boldsymbol{h}_t^{l-1})) \tag{5}$$

The denoised embedding is obtained directly from the hidden state representation for $\boldsymbol{e}_n$ at the final layer:

$$\boldsymbol{e}_d = \boldsymbol{h}_0^L \tag{6}$$

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

Table 1: Accuracies on downstream tasks for BERT

#### 4.3.2 COMPARISON WITH BASELINE

In Figure 4.3.2, we assess and compare the performance of three model families using three distinct metrics: AUC (Area Under Curve) for overall efficacy, MSE (Mean Squared Error) for model loss, and Cosine Similarity between the initial and recovered embeddings to gauge the effectiveness of our denoising mechanism. Against each metric, we compare our techniques with the baseline method TokEmbPriv Qu et al. (2021b).

For the three model families, we selected three distinct eta levels for experimentation, given the varying noise tolerance of each model. Specifically, for the Bert models, we set eta to 50, 100, and 500; for the GPT models, the values were 1, 100, and 1000; and for the T5 models, we chose 0.1, 1, and 10. For each model family, a representative task was selected. Our observations indicate that our method outperforms the baseline in terms of AUC in roughly 77% of the tested combinations, highlighting our model's ability to retain utility. Moreover, in approximately 90% of the combinations, we noted substantially reduced MSE values and enhanced Cosine Similarity scores. This suggests our technique's proficiency in restoring the original attributes of the noised embedding after procuring the perturbed results from the server.

| eta | T5 Small (60m) | | | | T5 Base (220m) | | | | T5 Large (770m) | | | |
|------|------|------|------|------|------|------|------|------|------|------|------|------|
|      | 0.001 | 0.01 | 1 | $\infty$ | 0.001 | 0.01 | 1 | $\infty$ | 0.001 | 0.01 | 1 | $\infty$ |
| CoLA | 0.69 | 0.69 | 0.69 | 0.71 | 0.70 | 0.69 | 0.69 | 0.73 | 0.69 | 0.69 | 0.69 | 0.74 |
| QQP  | 0.68 | 0.69 | 0.68 | 0.71 | 0.63 | 0.69 | 0.69 | 0.72 | 0.63 | 0.62 | 0.70 | 0.71 |
| MRPC | 0.68 | 0.68 | 0.68 | 0.69 | 0.68 | 0.68 | 0.68 | 0.71 | 0.68 | 0.68 | 0.68 | 0.71 |
| RTE  | 0.51 | 0.52 | 0.54 | 0.58 | 0.52 | 0.54 | 0.55 | 0.56 | 0.55 | 0.56 | 0.53 | 0.58 |

Table 2: Accuracies on downstream tasks for T5

| | GPT2 Small (120m) | | | GPT2 Medium (345m) | | | GPT2 large (774m) | | | GPT2 Xlarge (1.5b) | |
|---|---|---|---|---|---|---|---|---|---|---|---|
| eta | 1 | 100 | $\infty$ | 1 | 100 | $\infty$ | 1 | 100 | $\infty$ | 100 | $\infty$ |
| CoLA | 0.688 | 0.684 | 0.709 | 0.688 | 0.692 | 0.728 | 0.691 | 0.69 | 0.724 | 0.690 | 0.766 |
| QQP | 0.617 | 0.600 | 0.716 | 0.626 | 0.632 | 0.711 | 0.615 | 0.631 | 0.721 | 0.592 | 0.721 |
| MRPC | 0.688 | 0.681 | 0.720 | 0.684 | 0.686 | 0.710 | 0.674 | 0.684 | 0.701 | 0.676 | 0.705 |
| RTE | 0.535 | 0.549 | 0.579 | 0.556 | 0.567 | 0.583 | 0.567 | 0.596 | 0.582 | 0.574 | 0.592 |

Table 3: Accuracies on downstream tasks for GPT2

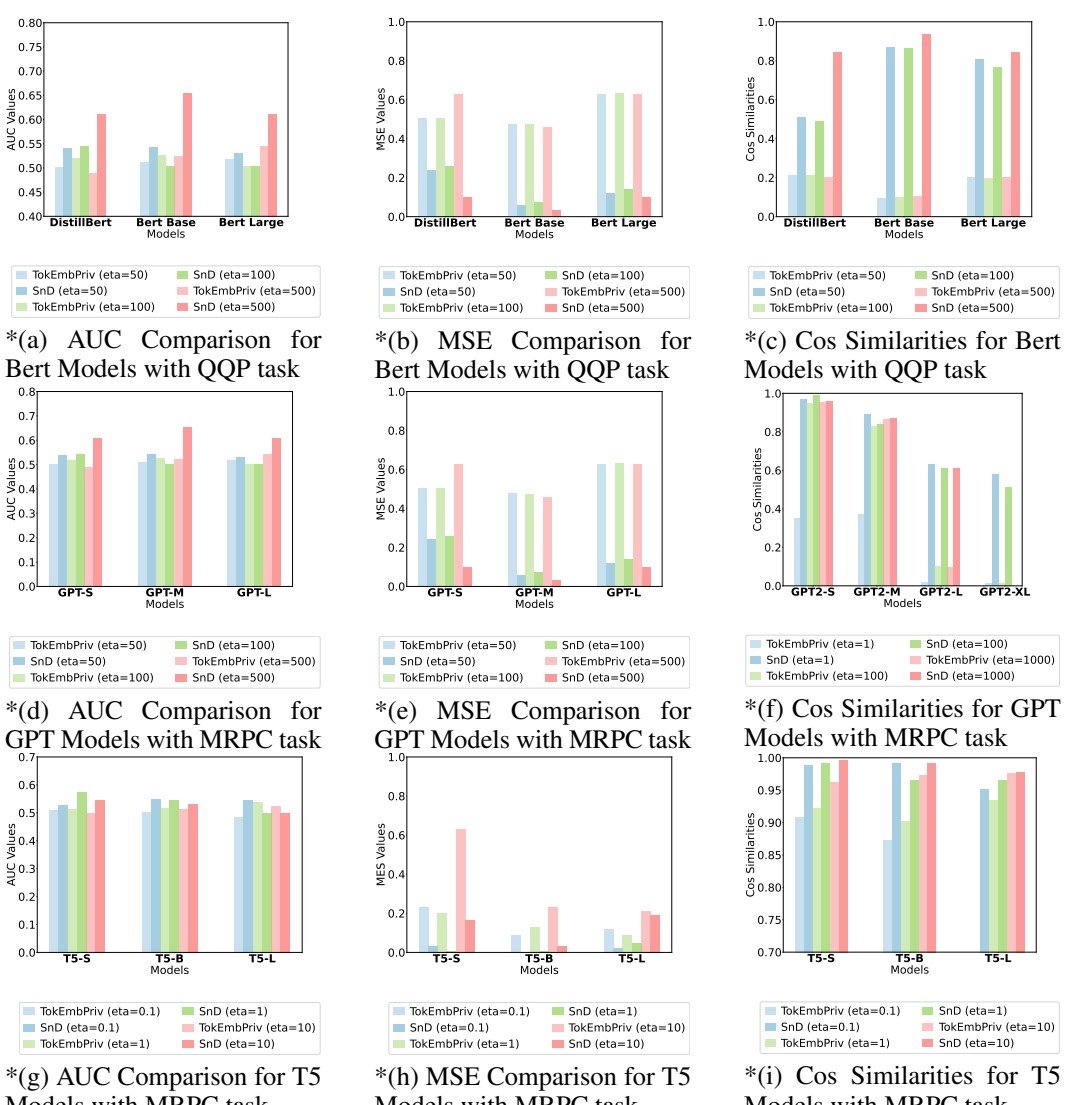

*(a) AUC Comparison for Bert Models with QQP task

*(b) MSE Comparison for Bert Models with QQP task

*(c) Cos Similarities for Bert Models with QQP task

*(d) AUC Comparison for GPT Models with MRPC task

*(e) MSE Comparison for GPT Models with MRPC task

*(f) Cos Similarities for GPT Models with MRPC task

*(g) AUC Comparison for T5 Models with MRPC task

*(h) MSE Comparison for T5 Models with MRPC task

*(i) Cos Similarities for T5 Models with MRPC task

Figure 3: Baseline performance comparison of 3 model families under multiple eta levels

# 5 DISCUSSION AND FUTURE WORK

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

Table 4: Hyperparameters of denoise models