# OpenReview forum: "Split-and-Denoise: Protect large language model inference with local differential privacy"
_ICLR.cc/2024/Conference — Submitted to ICLR 2024_

### Official Review · Reviewer_NdAW · 2023-10-31

**Soundness:** 3 good
**Presentation:** 3 good
**Contribution:** 3 good
**Rating:** 8
**Confidence:** 3

**Summary:**

This paper introduces a framework for achieving private Large Language Model (LLM) inference. The framework combines dx-privacy-based relaxed local Differential Privacy (DP), U-shape split inference, and a client-side denoiser based on the Transformer architecture. This innovative approach delivers efficient computation, privacy guarantees, and strong utility for private LLM inference. The paper demonstrates the effectiveness of the proposed framework using four datasets and three different architectural configurations. The results illustrate that the model maintains high utility while preserving a substantial privacy budget, with minimal computational complexity overhead.

**Strengths:**

1. The proposed work intelligently integrates split inference, relaxed local DP, and client-side post-processing with a denoiser.
2. The location of the denoising model at the client side, which leverages knowledge of the noise level, is crucial for effective denoising.
3. The paper provides a well-analyzed discussion of complexity.
4. The potential for several extension works pointed out in this work is interesting.

**Weaknesses:**

1. Recent work by "Mattern et al. - The Limits of Word Level Differential Privacy" has pointed out limitations of dx-privacy. The author should address these weaknesses and concerns related to dx-privacy in the paper.
2. The denoise model is pre-trained and known by the server. Could the server utilize the Denoise model to obtain less noisy user embeddings and potentially compromise privacy? This issue should be explored.
3. The impact of knowing the noise level or not should be more thoroughly analyzed.
4. The paper does not adequately demonstrate the privacy aspects regarding textual input. There is a lack of evidence that the proposed method can effectively defend against potential inversion attacks compared to an unprotected scheme.
5. The paper creates some confusion due to the mixed usage of "eta" in both its symbol and word forms.
6. There is a typographical error: "under varying η in table 4.3.1, 4.3.1 and 4.3.1."

**Questions:**

1. How does the Performance on selected benchmarks compare to SoTA?

2. What does eta mean for privacy? Why can it be set differently for GPT, Bert, and T5 models?

3. Can this be extended to training (finetuning)?

---

> ### Author Response · Authors · 2023-11-19
> **Response to Reviewer NdAW**
>
> Dear Reviewer NdAW,
>
> We sincerely appreciate your comment. Hope our additional experiments and clarifications could address your concerns.
>
> **Q1. Privacy concerns of denoise model.**
>
> The training of denoise model relies solely on the public dataset, and thus the denoise model is irrelavant to the private data. We added the clarification at the end of Section 3.4. Furthermore, the input to the denoise model requires the knowledge of specific noise matrix, which is not accessible by the server.
>
> **Q2. The impact of knowing the noise level or not**
>
> We added theoretical (see Section 3.4) and empirical studies (see Section A.10) to assess the impact of noise level awareness. For theoretical analysis, we showed that the denoising ability is inherently in tension with the level of privacy protection and provided a lower bound on the MSE of denoised results. For empirical analysis, we run the experiment that deploys the denoise model on server side using only privatized token representations as input. The results show that SnD significantly outperforms the server-side denoise method by over 10% in almost all cases for MRPC task.
>
> **Q3. Empirical evaluation of privacy protection & different $\eta$s**
>
> Refering to general response to Q1, we added experiment considering the inference attacks in Section 4.3 and 4.4.3. We found that the three classes of models show different robustness exhibit distinct robustness against inference attacks, and thus the privacy budget varies across the models. The $\eta$ is selected that meets the following two criteria simultaneously: (a) the accuracy for embedding inversion attack is below 0.01, and (b) the accuracy for attribute inference is below 0.53 (the accuracy for random guessing is around 0.5).
>
> Regarding the interpretation of $\eta$, it can be viewed as a parameter controlling the indistinguity between the distribution of the randomized mechnisms with two possible inputs $x$, $x'$. The log-likelihood ratio of observing any particular output $y$ given two inputs $x$, $x'$ is bounded by $\eta d(x, x')$, where $d(x, x')$ is the distance metric of the two inputs. In this paper, we adopted the Euclidean norm as the distance metric.
>
> **Q4. Typo errors and expression issues**
>
> Thanks for pointing out. We corrected the typos and used consistent $\eta$ in the paper.
>
> **Q5. Extension to training/finetuning**
>
> That’s a good question. While the privatization method described in our paper can be utilized during the training stage, directly applying the denoise model to the training/finetuning stage would pose a significant challenge. It would be another field of research to protect LLM training with local differential privacy while maintaining utility.
>
> **Q6. Limitations of $d_{\chi}$-differential privacy**
>
> The method proposed by "Mattern et al. - The Limits of Word Level Differential Privacy" couldn’t be directly applied to our framework, as it involves repharasing the sentence and thus is not compatible with the denoise model. Indeed, $d_{\chi}$-DP has been used frequently for text inputs in recent research [1]. Below we present some discussions on the weakness of the $d_{\chi}$-DP bound:
>
> -	Linear growth of privacy budget with token length. In the experiment, we limited the input length within 512 tokens. We also conducted inference attack on the privatized token representations to verify the privacy protection capability. Furthermore, in Section A.11, we show that our approach retains the performance even under extremely low privacy budgets, indicating that the user could set a very low privacy budge in case of possible leakage for long text inputs.
> -	Lack of syntactic changes. The syntactic properties of the original text could be hided when the privacy budget is set to a very low level.
> -	Limitation for usefulness of the text data on local DP. To preserve the utility of the subsequent output, we deploy a denoise model on user side to conduct error correction.
>
> [1] Feyisetan, O., Balle, B., Drake, T., & Diethe, T. (2020, January). Privacy-and utility-preserving textual analysis via calibrated multivariate perturbations. In Proceedings of the 13th international conference on web search and data mining (pp. 178-186).

---

> > ### Comment · Reviewer_NdAW · 2023-11-20
> > **Thanks for the response**
> >
> > Thank the authors for addressing my comments. You did an impressive work in the rebuttal and the revised paper addressed most of my concerns. I will increase the score.
> >
> > I just have one suggestion regarding "The denoise model is pre-trained and known by the server. Could the server utilize the Denoise model to obtain less noisy user embeddings and potentially compromise privacy?" - As indicated by A.10 Figure 6, the Denoise model on the server side seems not effective only slightly over 50% on a binary indication MRPC task - which means it would not raise a privacy problem since the server cannot crack it. This important message that should be included. Also, the title "Impact of user-side denoise model:" is not correct, the correct one is "Impact of server-side denoise model".
> >
> > **To other reviewers:** Please check the revised version if you have not done so, most of you raised questions on practicality and evaluation. I think the authors addressed them quite well.

---

> ### Author Response · Authors · 2023-11-21
> **Thanks for the reply**
>
> Dear Reviewer NdAW,
>
> Thanks a lot for acknowledging and valuing our efforts. We sincerely appreciate your support in our work. We have updated the expression in Section A.10 and underscored the server's inability to infer private information.
>
> Best regards,
>
> Authors of Submission 5188

---

### Official Review · Reviewer_C4Af · 2023-10-31

**Soundness:** 2 fair
**Presentation:** 3 good
**Contribution:** 2 fair
**Rating:** 3
**Confidence:** 3

**Summary:**

The paper proposes an approach of implementing local differential privacy to protect LLM inference. Specifically, a framework, SnD, is designed to enable the client to introduce noise prior to transmitting the embeddings to the server, and then denoise the output embeddings received from server, in a Embedding-as-a-Service scenario.

**Strengths:**

+ The study focuses on an interesting and important topic, the privacy in LLM.
+ The Embedding-as-a-Service business model is well-defined.

**Weaknesses:**

- The overhead of proposed approach is not clear

My first concern pertains to the overhead introduced by the local encoder. While a basic complexity analysis is presented in Section 3.5, I would recommend a more comprehensive evaluation involving specific experiments. Such an evaluation is crucial to assessing the feasibility of the proposed approach. Furthermore, it remains unclear how the denoise model, pre-trained on the server, can adapt to varying noise levels determined by clients.

- Lack of details on baseline description

There is a lack of detail in the baseline description. The only information provided about the benchmark method is "where the token embeddings are perturbed by the user before sending them to the server". It would be beneficial to provide more information about TokEmbPriv and explain why this method was chosen as the sole benchmark in the evaluation. Additionally, it would be helpful to clarify why other methods, such as a vanilla model, were not considered in the comparison.

- The evaluation on denoise is somehow blur

The evaluation of the denoising process is somewhat unclear. Please explicitly state and explain why a higher Cosine Similarity score between the initial and recovered embeddings is indicative of better performance. Also, please provide a clear definition of what "initial embeddings" refer to. If they pertain to the embeddings input to the denoise module, i.e., the "output noised results" in Figure 1, it currently reads as though having fewer differences between the initial and recovered embeddings is preferred, which might lead to less noise added. Additionally, the evaluation appears to focus solely on accuracy and does not provide results on privacy protection. Given that the primary motivation of the study is addressing the "unaddressed risk of privacy leakage," the privacy performance of the proposed approach is expected and should be included in the evaluation.

**Questions:**

1. What is the overhead of the proposed approach?
2. Why TokEmbPriv is chosen as the sole benchmark in the evaluation?
3. What is the privacy performance of the proposed method?

---

> ### Author Response · Authors · 2023-11-19
> **Response to Reviewer C4Af**
>
> Dear Reviewer C4Af,
>
> Thanks for your comments. Hope our modifications and answers below could address your concerns.
>
> **Q1. The overhead of proposed approach**
>
> We added experiment in Appendix A.9 to evaluate overhead of SnD. Refer to the general response to Q3.
>
> **Q2. Denoise model for varying $\eta$ levels**
>
> In Appendix A.6 we added the specification of training the denoise model. Simply speaking, we devide the $\eta$ in to three groups according to the correlation between the privatized and plain token representation tokens, and train a seperate denoise model for each group. During inference, the user could choose the corresponding denoise model based on their privacy requirement.
>
> **Q3. Baseline description & lack of baseline**
>
> Refering to the answer to Q2 in general response, we added two baselines in the experiments. We also present more illustration of the baseline methods in “prior works” and Appendix A.1. The accuracy performance for a vanilla model is presented in the case of $\eta=\infty$ in table 1, 2, and 3, where no noises are added and denoise model is not deployed on user side.
>
> **Q4. Privacy performance of the proposed method**
>
> Refering to general response to Q1, we added experiment considering the inference attacks in Section 4.3 and 4.4.3.
>
> **Q5. Interpretation of Cosine Similarity**
>
> The cosine similarity measures the similarity between the clean and denoised embeddings in the case of SnD (or privatized embedding in the case of text2text and tokEmbPriv). Clean embedding refers to the output embedding computed from the plain token embeddings, and denoised embedding refers to the output embedding derived from the privatized token embeddings and transformed by the denoise model. A higher cosine similarity means that the denoised embedding is more similar to the original clean embeddings and thus is prefered. We make the expression clearer in Section A.8.

---

> ### Author Response · Authors · 2023-11-21
> **Look forward to your reply**
>
> Dear Reviewer C4Af,
>
> As the discussion period comes to a conclusion, we are eager to understand if our response and revision have addressed your concerns. Your further insights would be valuable as we endeavor to enhance our submission in these final days. We sincerely appreciate your time and guidance.
>
> Best Regards,
>
> Authors of Submission 5188

---

> ### Author Response · Authors · 2023-11-22
> **Follow up on Q3**
>
> Dear Reviewer C4Af,
>
> In the latest version, we have added the description and implementation details for the baseline methods in Section A.6.5. As the deadline of ICLR rebuttal period is approaching, we look forward to hearing from you and would be pleased to address any remaining concerns that you may still have.
>
> Best Regards,
>
> Authors of Submission 5188

---

### Official Review · Reviewer_xFRn · 2023-11-01

**Soundness:** 2 fair
**Presentation:** 1 poor
**Contribution:** 2 fair
**Rating:** 3
**Confidence:** 2

**Summary:**

The authors introduce Split-N-Denoise (SnD), a privacy-preserving scheme for LLMs.  SnD makes use of split learning, wherein the network is chopped in two and each half is distributed to the client and server, respectively.  SnD splits at the embedding layer, where local differential privacy (LDP) is applied to user embedding vectors for obfuscation.  Along with the embedding layer, the client side also contains a trained denoiser which, after the output embedding vectors are transmitted from the server, is used to denoise the response.  Several experiments are conducted demonstrating the efficacy of this work, compared to another method designed specifically for BERT (i.e., TokEmbPriv from Qu et al, 2021), across several models (BERT/DistillBert, T5, and GPT-2) for three classification tasks.

**Strengths:**

The problem of how to preserve-privacy in embedding-as-a-service applications is important with the influx of interest in LLMs.  Furthermore, the authors use of LPD provides certifiable privacy and, in the case of split learning, LDP is a very nontrivial task.  The presented framework makes sense in this context, although the overall approach could be better motivated.  For instance, why split at the embedding layer?  Splitting at this particular layer is the most unsecure, even for a simple man in the middle attack (given a pretrained foundation model).  However, splitting at any other layer and transmitting gradients naturally allows for federated learning strategies and the avoidance of DP altogether.  Furthermore, multi-party computation (MPC) methods do not have the encountered problem of trying to use DP with split learning (and properly denoising transmitted data).  A contrast and discussion of these various approaches is warranted.  Please see the following for a recent privacy-preserving MPC method:
Knott, Brian, et al. "Crypten: Secure multi-party computation meets machine learning." Advances in Neural Information Processing Systems 34 (2021): 4961-4973

**Weaknesses:**

# Autoregressive LLM concerns regarding the (only) use of GPT-2 and evaluation

The featured classification tasks do well to test the performance of BERT and T5.  However, they do not relevantly test the performance of GPT-2; in practice, GPT-2 does not perform well on classification tasks, which would ideally be handled by more appropriate encoder (e.g., BERT) or encoder-decoder (e.g., T5) architectures.  More relevant generative metrics, such as sentence completion (e.g., via HellaSwag), next-word-completion (e.g., via the lambada dataset), or even perplexity are required to show that GPT-2 performance is maintained.  Furthermore, - GPT-2 itself is an older model and was trained in a much less complicated manner compared to recently released (instruction-tuned) models, such as LLaMa-2, Falcon, Mistral, etc..  Testing on a more recent architecture is important to show the efficacy of this approach, to show that the more sophisticated pretrained knowledge (which naturally contains many more learned modalities/instructions compared to GPT-2) still behaves as expected, and to show different architectural choices are unaffected by the introduced noise (e.g., LLaMa uses RMSNorm vs GPT-2's Layernorm, which, relevant to the presented work, directly impacts how these models deal with noise variation in the data).

# Lack of: (a) comparison to other relevant privacy-preserving methods, (b) evidence against competitors infeasibility

Beyond TokEmbPriv, comparison is lacking to other relevant benchmark competitors.  E.g., due to the similarity of both approaches, a comparison to RAPT (Li et al. (2023)) is necessary.  Furethermore, for homomorphic encryption approaches, the authors claim:
> Cryptographic typically employs homomorphic encryption (HE) to compute the inference result of the users’ encrypted input. Unfortunately, the application of cryptographic technique is constrained by the significant computation overhead of cryptographic operations, especially on large transformer models.

To assert this claim, the proposed method and reference HE methods should be compared in terms of both accuracy, privacy-preserving ability, and wall-clock time.

# Inaccurate claims
> To the best of our knowledge, this paper represents the pioneering effort in protect user’s privacy during LLM inference with strong privacy guarantee. Existing research focuses on the privacy-preserving pre-training and fine-tuning for LLM, while few studies pay attention to the privacy concerns at inference stage, especially on privatizing user’s input to guarantee DP.

The homomorphic encrpytion work by Liu & Liu (2023); Chen et al. (2022) (cited in the paper) protects user privacy during inference, please withdraw or appropriately revise this claim.

Furthermore, the description of RAPT (which itself offers LDP for privacy-preserving LLM inference) is inaccurate:
> An alternative strategy might entail input text perturbation via textto-text privatization or synthetic data generation, preserving high-dimensional features while altering human-perceivable sequences Li et al. (2023). Specifically, the text\
-to-text privatization projects text into a high-dimensional vector space with a pre-determined word embedding model,adding
carefully calibrated noise to the vector representation, and then reconvert it to obtain the perturbed
text Feyisetan et al. (2019); Qu et al. (2021a). Yet, the mere application of this technique during
inference does not guarantee a satisfactory balance between privacy and utility

This is a misleading description of the RAPT method from Li et al. (2023); RAPT performs LDP to user prompts (thus establishing certifiable privacy via DP).  In order to recover performance on the noisy data, RAPT employs prompt tuning to efficiently fine-tune a server-side model capable of performing inference on the LDP data.  This approach is extremely similar to the proposed split-and-denoise framework.  Please clearly describe RAPT and contrast it to SnP, while also benchmarking against it as a relevant competitor.

> Split learning Gupta & Raskar (2018); Vepakomma et al. (2018) ... DP is employed to mitigate privacy leakage by injecting noises into the IRs before sharing with the server.

Neither of the two papers use DP to certifiably protect the data.  Please cite another multi-party computation paper (where intermediate gradients are distributed across networks) which uses DP

# Lack of demonstrated efficacy against attacks

It is necessasry to prove the privacy-preserving capabilities of the presented approach, e.g., simulate eavesdropping attacks, which collect intercepted embedding vectors (during transmission to the server) for malicious actions (e.g., embedding inversion and attribute inference attacks).  Please see Li et al. (2023) for examples and more details.

**Questions:**

> We design a novel denoising method deployed on user side. In this approach, a denoise
model is pre-trained on server side using public dataset and synthetic noises. Subsequently,
this trained model is deployed on the user side, where it leverages the specific noise levels
and raw IRs provided by the user to enhance the embeddings.

Can the authors comment on how this necessarily opens up a large security hole? I.e., the data has noise injected to protect it in the event of interception by a bad actor.  However, this scheme requires the transmission of the actual denoising layer, which itself may be intercepted.  Due to the need for model refreshes, this problem is non-trivial.

> Split learning is a novel privacy-preserving approach in distributed learning

Please remove "novel", as it not being introduced in the presented work.

The benchmarked method, "TokEmbPriv," requires significantly more discussion during the background and previous work sections.

The paper requires an editing pass.

---

> ### Author Response · Authors · 2023-11-19
> **Response to Reviewer xFRn**
>
> Dear Reviewer xFRn,
>
> Thanks for your suggestions. We have enhanced our paper based on your comment, and hope that will address your concerns.
>
> **Q1. Testing of GPT2 on classification task**
>
> The autoregressive GPT model has been evaluated on classification tasks in recent research [1]. Furthermore, OpenAI has released the Embedding-as-a-Service (EaaS) for downstream tasks including classification [2], where GPT2 and 3 are provided as the underlying models. Therefore, it’s of pratical concerns to test the performance of GPT2 on the downstream tasks given by the paper. Meanwhile, we are conducting experiment of GPT2 on HellaSwag and lambada dataset, but we haven't completed it due to computer resource and time limit. We will add the results to the paper once the experiments are done.
>
> **Q2. Comparison to other baselines**
>
> We added the comparison of SnD with more privacy-preserving LLM inference methods. In Section 4.1 and 4.4.2, we compare the auc of SnD on downstream tasks with text2text and RAPT. Benefiting from the denoise model, our method results in AUC higher than the three baselines with an average over 10\%.
>
> In Section A.9, we evaluated the computation and communication overhead for two encryption-based method [3] [4], and found that our method results in thousands of speedup for the SOTA HE algorithms. The privacy comparison between PCFT and SnD can be found in Section A.1, where PCFT adopts the security definition in multiparty computation while SnD protect the data with LDP.
>
> **Q3. Difference between RAPT and SnD**
>
> In Section A.1, we illustrated the distinction between RAPT and our framework. Though RAPT employs a reconstruction head to improve the robustness against noises during prompt-tuning stage, *the module only reconstruct the random tokens and doesn’t perform any direct denoise either on the privatized input or the corresponding representations/activations for the private token.* Furthermore, *the reconstruction head is discarded during the inference stage, rendering no denoise mechanism for LLM inference.* In constrast, our work performs explicit local denoise for the output embedding on the user-side, which demonstrates better performance.
>
> **Q4. Privacy-preserving capabilities**
>
> Refer to the general response of Q1, we added experiment considering the inference attacks in Section 4.3 and 4.4.3. Such attack could be carried out by an eavesdropper or the curious server.
>
> **Q5. Privacy concerns for denoising layer**
>
> The training of denoise model relies only on the public dataset, and thus the denoise layer release no information about the private data. We added the clarification at the end of Section 3.4.
>
> **Q6. Inaccurate claims**
>
> We rewrited the claims and highlighted them in blue.
>
> [1] Radford, A., Narasimhan, K., Salimans, T., & Sutskever, I. (2018). Improving language understanding by generative pre-training.
>
> [2] https://platform.openai.com/docs/guides/embeddings/what-are-embeddings
>
> [3] Liu, X., & Liu, Z. (2023). LLMs Can Understand Encrypted Prompt: Towards Privacy-Computing Friendly Transformers. arXiv preprint arXiv:2305.18396.
>
> [4] Hao, M., Li, H., Chen, H., Xing, P., Xu, G., & Zhang, T. (2022). Iron: Private inference on transformers. Advances in Neural Information Processing Systems, 35, 15718-15731.

---

> > ### Comment · Reviewer_xFRn · 2023-11-21
> > **Response to rebuttal**
> >
> > Thank you to the authors for the new experiments and details.  I've looked at both your reply and changes to the paper.
> >
> > ### Q1. Testing of GPT2 on classification task
> >
> > > The autoregressive GPT model has been evaluated on classification tasks in recent research [1]. Furthermore, OpenAI has released the Embedding-as-a-Service (EaaS) for downstream tasks including classification [2], where GPT2 and 3 are provided as the underlying models. Therefore, it’s of pratical concerns to test the performance of GPT2 on the downstream tasks given by the paper. Meanwhile, we are conducting experiment of GPT2 on HellaSwag and lambada dataset, but we haven't completed it due to computer resource and time limit. We will add the results to the paper once the experiments are done.?
> >
> > It is important to note that the EaaS solution by OpenAI is not the same as evaluating/reporting GPT-2 performance on classification tasks.  EaaS uses a given model's embedding matrix as a feature store (i.e., vector DB) to be passed onto other models for different downstream tasks (e.g., OpenAI gives an example of training regressors/classifiers from sklearn using embedding vectors).  This is inherently not the same as training an autoregressive LLM, such as GPT-2, to perform classification tasks.  I look forward to the HellaSwag and Lambada results.
> >
> > Thank you for including both more attack use cases and comparison to more competitors.  Can you please add more implementation details on how competitors were run?  E.g., for all approaches which use DP, what privacy thresholds were used?
> >  How were the methods implemented (e.g., software version, when avaiable)?  How were hyperparameters chosen to fairly reflect all considered algorithms performance?  Note that RAPT uses the reconstruction head during training to force that the LLM can naturally decode the LDP tokens; it is not surprising the reconstruction head may thus be discarded during inference, and the ability of the LLM to naturally decode the LDP tokens is the entire selling point of RAPT.  Furthermore, some discussion on the limitations of TokEmbPriv is still warranted; this method was specifically designed and tested for BERT models (this limitation can be described in the authors' favor), why would it be expected to perform well applied to GPT-2?  Please note these limitations/explanations of performance within the text.

---

> ### Author Response · Authors · 2023-11-21
> **Look forward to your reply**
>
> Dear Reviewer xFRn,
>
> As the discussion period comes to a conclusion, we are eager to understand if our response and revision have addressed your concerns. Your further insights would be valuable as we endeavor to enhance our submission in these final days. We sincerely appreciate your time and guidance.
>
> Best Regards,
>
> Authors of Submission 5188

---

> ### Author Response · Authors · 2023-11-22
> **Response to Reviewer xFRn's Reply**
>
> Dear Reviewer xFRn,
>
> Thanks for your reply. I have uploaded a revised version of paper based on your suggestion. Hope the revision and answers below could address your concern.
>
> **Q1. Testing of GPT2 on classification task.**
>
> Our EaaS model works similar to OpenAI's solution. Instead of finetuning GPT2 on the classification task, we employ the LLM model to extract embeddings and use the embeddings as the feature score to train a separate classification model on the downstream task. For more details refer to A.5. We also mention this point in Table 7 of Section A.1.
>
> **Q2. Implementation details on how competitors were run**
>
> Thanks for the suggestion. We have added the implementation details in section A.6.1 and A.6.5. Noted that all DP methods utilizes the same $d_{\chi}$-privacy with Euclidean distance as metrics.
>
> **Q3. RAPT's reconstruction head**
>
> We understand that the reconstruction head help the model to better understand the privatized input. Since the reconstruction module works solely on decoding random tokens, we note that the precise mechanism by which the LLM learns to decode the privatized tokens remains unclear. We updated the expression to be more precise in Section A.1.
>
> **Q4. Discussion on the limitations of TokEmbPriv**
>
> Thanks for the suggestion. We included the discussions on the limitation of TokEmbPriv and Text2Text in Section 4.4.2. The lack of  denoise mechanism and unbounded noise support in TokenEmbPriv might lead to its poorer performance in downstream task.

---

### Official Review · Reviewer_4SyG · 2023-11-05

**Soundness:** 2 fair
**Presentation:** 2 fair
**Contribution:** 2 fair
**Rating:** 5
**Confidence:** 4

**Summary:**

This paper focuses on the problem of privacy-preserving LLM inference in a setting where clients input text and a server holds the model. To address the privacy risk of direct transmission of clients' text to server, this paper splits an LLM between clients and a server such that:

1) Local computation: the client performs affordable computation locally to obtain intermediate results. In particular, only the token embedding is done on the client side to keep the computational cost low for clients.

2) Privatising clients submissions: Differential privacy is employed to mitigate privacy leakage by injecting noises into the embedding before sharing with the server. In particular, each client adds noise to their embedding prior to sending them to the server to protect the privacy of clients while doing LLM inference;

3) Server-side computation: the server receives the noisy embedding, and performs the rest of the computations of the LLM and returns the noisy output to the client.

4) Client-side denoising: Each client performs denoising to improve the utility of the output. The denoise model is pre-trained on the server side using public datasets and synthetic noises, and subsequently shared with the client.

**Strengths:**

1) As opposed to the server-side denoising that has been used in the existing work, this paper performs the denoising at the client side to leverage the knowledge of noise levels and raw embedding.

2) Evaluation of the proposed method through computing similarity between the clean and privatized embeddings, and performance on downstream tasks of sentence classification, pair similarity, Recognizing Textual Entailment

3) The problem of privacy-preserving LLM inference which is studied by this paper is an important problem as clients may input sensitive information, such as names, phones, and email addresses, that needs to be kept hidden from the service provider.

**Weaknesses:**

1) No evidence to support the practicality of the proposed method. This paper instantiates the denoising model as an L layer transformer-based model that receives the privatised token representations, noise matrix and noisy output computed by the server. Although this denoise model is pre-trained on the server side using public datasets and synthetic noises, each client needs to do the inference of this denoising model locally. Unfortunately, this paper does not empirically evaluate the memory and computational cost of this inference for clients. Therefore, it is not clear if this overhead is any smaller than running the whole LLM on the client side. This evaluation is necessary as one main claim of this paper is that the proposed method introduces only affordable local computations for clients.

2) No evidence demonstrating the privacy benefits of the proposed method. This paper lacks an empirical evaluation of the privacy leakage of the proposed method. This is particularly important as analytical privacy guarantees chosen by this paper are very loose for example see privacy budgets of 100, 500 and 1000 in the Tables provided in the experiment section.


3) Shallow discussion of results and not considering SOTA models.

4) Experimental choices including hyperparameters and privacy budgets are not justified/studied and they are not consistent across models. The only statement regarding the choice of the privacy budget that I can see in the paper is the following: "For the three model families, we selected three distinct eta levels for experimentation, given the varying noise tolerance of each model. Specifically, for the Bert models, we set eta to 50, 100, and 500; for the GPT models, the values were 1, 100, and 1000; and for the T5 models, we chose 0.1, 1, and 10."


5) There are many typos:
  1) Missing "." at the end of captions
  2) Consequently, As a result,: As --> as
  3) known as ”embedding as a service”: ”embedding --> ``embedding
  4) this paper represents the pioneering effort in protect user’s: protect --> protecting
  5) Fact Kotonya & Toni (2020),Daily Dialogue: ,Daily --> , Daily

**Questions:**

I would recommend reporting the computational costs and privacy benefits of the proposed method. Please see my first and second concerns in the weakness box.

---

> ### Author Response · Authors · 2023-11-19
> **Response to Reviewer 4SyG**
>
> Dear Reviewer 4SyG,
>
> Thanks for your comments. Hope the following response and our additional experiments could address your concerns.
>
> **Q1. Evidence to support the practicality of the proposed method.**
>
> We added experiment in Appendix A.9 to compare the computation and memory cost of user-side inference of SnD with the case where the whole model is downloaded. The computation time and memory cost saves by SnD ranges from, respectively, 78% to 96% and 65% to 94%, across the 10 models. The saving is more significant for models of large size, such as GPT2 Xlarge.
>
> **Q2. Evidence for the privacy benefits & consistence of privacy budget.**
>
> We added embedding inversion attack and attribute inference attacks to verify the privacy protection cabilities under different $\eta$s in Section 4.4.3 and 4.3. We found that the three classes of models show different robustness exhibit distinct robustness against inference attacks, and thus the privacy budget varies across the models. The $\eta$ is selected that meets the following two criteria simultaneously: (a) the accuracy for embedding inversion attack is below 0.01, and (b) the accuracy for attribute inference is below 0.53 (the accuracy for random guessing is around 0.5). Noted that we have updated the privacy budget for GPT2 to be 1, 50, 100. We also added evaluation for the performance of extremely low privacy level in Appendix A.11.
>
> **Q3.  Shallow discussion of results.**
>
> We enhance the discussion in the Section 4. Compared with the vanilla model, Bert, T5, and GPT models yield average model losses of 4.31\%, 4.48\%, and 5.25\%, respectively. In addition, SnD outperforms the baselines by an average of over 10\% benefiting from the denoise model. More details are presented in our paper.
>
> **Q4. Typos**
>
> We have corrected the typos and conducted grammer check on the paper.

---

> > ### Comment · Reviewer_4SyG · 2023-11-21
> > **Reviewer follow up on Q1**
> >
> > Thanks for evaluating the overhead of your method. I have some questions/concerns regarding this evaluation and your comparison:
> > 1. How do you measure the computation costs and what is its unit in Table 12?
> >
> > 2. Why are the communication costs of encryption-based methods in Table 12 higher than your method?
> >
> > 3. Do clients share their encrypted data with the server in encryption-based methods? If so, how big is the sizeof input in comparison to the embedding which is shared by your method?
> >
> > 4. Also Figure 5 confirms my concern that the computation costs of your method is high and very close to the computation costs of running the whole LLM locally on the user's device. The y-axis is in seconds I would not show it with the steps size of $10^{-2}$ to not be misleading please.

---

> > > ### Comment · Reviewer_4SyG · 2023-11-21
> > > **Reviewer follow up on Q2**
> > >
> > > Thanks for empirically evaluating the privacy leakage of the proposed method. However, it is very hard to understand this experiment as
> > > 1. The description of your embedding inversion attack is missing.
> > > 2. How strong is your attack? Is it an existing SOTA attack or is it something that you designed.
> > > 2. Figure 2 misses labels for its x-axis and y-axis.
> > > 3. It is not clear if the attack is performed on the embedding obtained by your method or not?
> > > 4. Figure 2 misses a discussion.

---

> ### Author Response · Authors · 2023-11-21
> **Look forward to your reply**
>
> Dear Reviewer 4SyG,
>
> As the discussion period comes to a conclusion, we are eager to understand if our response and revision have addressed your concerns. Your further insights would be valuable as we endeavor to enhance our submission in these final days. We sincerely appreciate your time and guidance.
>
> Best Regards,
>
> Authors of Submission 5188

---

> ### Author Response · Authors · 2023-11-21
> **Response to Reviewer 4SyG's Reply**
>
> Dear Reviewer 4SyG,
>
> Thanks for your reply. We uploaded a revised version based on your suggestions. Hope our further clarification and revisions could address your concern.
>
> **Q1. How do you measure the computation costs and what is its unit in Table 12?**
>
> The computation costs are in second, which is indicated by the caption of Table 12.
>
> **Q2. Communication costs of encryption-based methods**
>
> The high communication cost is caused by the multiparty computation techniques adopted by the methods. Instead of only uploading the encrypted data to the server, the client and server transmit the secret shares as well as encrypted versions of intermediate values obtained from a series operations in the LLM. For more details refer to the secure computation protocols in [1] and [2]. I included this point in A.9.
>
> **Q3. Figure 5 confirms my concern that the computation costs of your method is high and very close to the computation costs of running the whole LLM locally on the user's device.**
>
> We are afraid that there might be some misunderstanding. It's important to note that our results reveal significant computation benifit of SnD compared with whole LLM inference. According to the Figure, SnD saves the computation cost by an average of over 85%, and in particular 95.3% for GPT2 Xlarge, compared with running the whole LLM locally. We also added more explanations in the caption of figure 5 to make it clearer.
>
> **Q4. The y-axis is in seconds I would not show it with the steps size of $10^{-2}$ to not be misleading please.**
>
> Could you make further illustration on this suggestions? Seems like there are some confusions on our side. We will strive to make the figure clearer based on your advice.
>
> **Q5. The description of your embedding inversion attack is missing**
>
> The description of embedding inversion attack is presented in Section 4.3. The server recover the tokens from the privatized token embedding based on the L_2 distance.
>
> **Q6. How strong is your attack? Is it an existing SOTA attack or is it something that you designed.**
>
> The attacks have been adopted in existing research to verify the effectiveness of SOTA methods [3][4]. We added the citation after each attack in Section 4.3.
>
> **Q7. Figure 2 misses labels for its x-axis and y-axis.**
>
> Thanks for pointing out. We added the labels in the latest version. The x-axis denotes the privacy budget and y-axis denotes attack accuracy.
>
> **Q8. It is not clear if the attack is performed on the embedding obtained by your method or not?**
>
> The attack is performed on the privatized token embedding obtained by our method. We included this point in Section 4.3.
>
> **Q9. Figure 2 misses a discussion.**
>
> The discussion for Figure 2 is presented in Section 4.4.3.
>
> [1] Liu, X., & Liu, Z. (2023). LLMs Can Understand Encrypted Prompt: Towards Privacy-Computing Friendly Transformers. arXiv preprint arXiv:2305.18396.
>
> [2] Hao, M., Li, H., Chen, H., Xing, P., Xu, G., & Zhang, T. (2022). Iron: Private inference on transformers. Advances in Neural Information Processing Systems, 35, 15718-15731.
>
> [3] Li, Y., Tan, Z., & Liu, Y. (2023). Privacy-preserving prompt tuning for large language model services. arXiv preprint arXiv:2305.06212.
>
> [4] Qu, C., Kong, W., Yang, L., Zhang, M., Bendersky, M., & Najork, M. (2021, October). Natural language understanding with privacy-preserving bert. In Proceedings of the 30th ACM International Conference on Information & Knowledge Management (pp. 1488-1497).

---

### Author Response · Authors · 2023-11-19
**General response to all reviewers**

Dear reviewers,

We sincerely appreciate your constructive comments and suggestions for our paper. Rest assured that we have carefully considered each of your questions and concerns. In our response, we will address all the specific queries raised by individual reviewers. Additionally, we have noticed some overlapping questions, and we plan to address them collectively in this general response section.

There are some updates in our papers (highlighted in blue):

1.	We added experiment for embedding inversion attack and attribute inference attack to verify the privacy protection under different $\eta$. (See Section 4.3 and 4.4.3)

2.	We added experiment for two baselines, text2text and RAPT. (See Section 4.1 and 4.4.2)

3.	We added the overhead analysis to verify the practibility of our method. (See Section A.9)

4.	Our method is slightly modified by incorporating a norm clipping procedure on the privatized token embedding, which leads to enhanced performance. (See Section 3.3 and A.9)

5.	We conduct more thorough theoretical and empirical analysis in terms of the benefits of denoising at user-side with the knowledge of noise levels. (See Section 3.4 and A.10)

6.	There are more discussions on the baselines in “prior works” and appendix. (See Section 2 and A.1)

**Q1. Privacy-preserving capabilities of the presented approach.**

Since only the privatized token representation is transmitted during the inference, we focus on the theoretical and empirical analysis in terms of the privatized token embeddings. For theoretical analysis, we show that the algorithm satisfy $\eta d_{\chi}$-DP. For empirical analysis, we conducted the experiments on embedding inversion attack and attribute inference attack under different $\eta$ levels [1].

**Q2. Comparison with SOTA models.**

We compare SnD with three baseline methods: (i) Token embedding privatization (TokEmbPriv) [2], (ii) Text-to-text privatization (Text2Text) [2][3], (iii) Privacy-Preserving Prompt Tuning (RAPT) [1]. Our approach demonstrates significant improvement in performence compared with the baselines.

**Q3. Evaluation of overhead.**

We evaluate the computation time, communication cost and memory requirement in Section A.9, which demonstrates the benefits in overhead compared with encryption-based methods, and local inference with the whole model.

[1] Li, Y., Tan, Z., & Liu, Y. (2023). Privacy-preserving prompt tuning for large language model services. arXiv preprint arXiv:2305.06212.

[2] Qu, C., Kong, W., Yang, L., Zhang, M., Bendersky, M., & Najork, M. (2021, October). Natural language understanding with privacy-preserving bert. In Proceedings of the 30th ACM International Conference on Information & Knowledge Management (pp. 1488-1497).

[3] Feyisetan, O., Balle, B., Drake, T., & Diethe, T. (2020, January). Privacy-and utility-preserving textual analysis via calibrated multivariate perturbations. In Proceedings of the 13th international conference on web search and data mining (pp. 178-186).

---

### Meta-Review · Area_Chair_Bn39 · 2023-12-08

**Metareview:**

This paper introduces a method for privacy protection in split inference using local DP. Given a language model, the client computes the input embedding and adds noise to the embedding before transmitting it to the server, with the noise magnitude calibrated to satisfy local metric-DP. Upon receiving the server output, the client uses a learned denoising model to denoise the output. Doing so greatly improves the model's utility without sacrificing the privacy guarantee.

Reviewers generally found the approach to be novel and interesting, especially the use of a denoising model to correct the server's output without sacrificing the privacy guarantee. The main weakness is its lack of comprehensive empirical privacy evaluation. Although the technique satisfies local metric-DP, the privacy parameter $\eta$ depends strongly on the semantic similarity in token embedding space, as well as how much the surrounding context reveals about a particular token. In fact, for the models that the authors tested, the suitable value of $\eta$ ranges from 100 in BERT/GPT to 1 in T5. Without resorting to a strong inference attack, it is very difficult to select the appropriate value of $\eta$, which significantly undermines the value of having a theoretical guarantee. The authors should consider strengthen their evaluation with well-designed inference attacks in order to address this critical weakness.

**Justification For Why Not Higher Score:**

The privacy parameter $\eta$ must be selected based on empirical evaluation of inference attacks, and the authors did not consider strong inference attacks for this purpose.

**Justification For Why Not Lower Score:**

N/A

---

### Decision · Program_Chairs · 2024-01-16

Reject